# ASIC1a is required for neuronal activation via low-intensity ultrasound stimulation in mouse brain

**Jormay Lim[1†], Hsiao-Hsin Tai[1†], Wei-Hao Liao[2†], Ya-Cherng Chu[1†], Chen-Ming Hao[1], Yueh-Chun Huang[1], Cheng-Han Lee[3], Shao-Shien Lin[4], Sherry Hsu[1], Ya-Chih Chien[3], Dar-Ming Lai[4], Wen-Shiang Chen[2], Chih-Cheng Chen[3]\*, Jaw-Lin Wang[1]\***

[1]Department of Biomedical Engineering, College of Medicine and College of Engineering, National Taiwan University, Taipei, Taiwan; [2]Department of Physical Medicine and Rehabilitation, National Taiwan Hospital University, Taipei, Taiwan; [3]Institute of Biomedical Sciences, Academia Sinica, Taipei, Taiwan; [4]Department of Surgery, National Taiwan Hospital University, Taipei, Taiwan

**ABSTRACT** Accumulating evidence has shown transcranial low-intensity ultrasound can be potentially a non-invasive neural modulation tool to treat brain diseases. However, the underlying mechanism remains elusive and the majority of studies on animal models applying rather high-intensity ultrasound that cannot be safely used in humans. Here, we showed low-intensity ultrasound was able to activate neurons in the mouse brain and repeated ultrasound stimulation resulted in adult neurogenesis in specific brain regions. In vitro calcium imaging studies showed that a specific ultrasound stimulation mode, which combined with both ultrasound-induced pressure and acoustic streaming mechanotransduction, is required to activate cultured cortical neurons. ASIC1a and cytoskeletal proteins were involved in the low-intensity ultrasound-mediated mechanotransduction and cultured neuron activation, which was inhibited by ASIC1a blockade and cytoskeleton-modified agents. In contrast, the inhibition of mechanical-sensitive channels involved in bilayer-model mechanotransduction like Piezo or TRP proteins did not repress the ultrasound-mediated neuronal activation as efficiently. The ASIC1a-mediated ultrasound effects in mouse brain such as immediate response of ERK phosphorylation and DCX marked neurogenesis were statistically significantly compromised by ASIC1a gene deletion. Collated data suggest that ASIC1a is the molecular determinant involved in the mechano-signaling of low-intensity ultrasound that modulates neural activation in mouse brain.

**\*For correspondence:**
chih@ibms.sinica.edu.tw (C-ChengC);
jlwang@ntu.edu.tw (J-LinW)

[†]These authors contributed equally to this work

**Competing interest:** The authors declare that no competing interests exist.

## Introduction

Transcranial ultrasound such as opening blood-brain barrier (BBB) (*Cammalleri et al., 2020*) for localized drug release and modulating neural activity (*Nicodemus et al., 2019*; *Legon et al., 2014*; *David et al., 2014*) has been used for therapeutic treatments of various brain diseases. Many in vivo animal experiments and human clinical trials (*Supplementary file 1*) proved the clinical potential of transcranial ultrasound stimulation. With the increased interest of this technique, the mechanisms underlying ultrasound-mediated neural modulation has also recently been learned. A study showed high-intensity transcranial ultrasound can elicit a startle-like motor response via an indirect auditory mechanism (*Sato et al., 2018*). Emerging sonogenetics in worm model also identified and engineered TRP-4 channels as a sensor for the ultrasound stimulus to activate neurons in living organisms at pressure level above 0.5 MPa (*Ibsen et al., 2015*), and ultrasound at 0.1 MPa acoustic pressure was found to activate neurons via Piezo one mechanosensitive ion channel (*Qiu et al., 2019*). Nonetheless, the

energy intensity or acoustic pressure of most clinical trials or basic researches used for BBB opening or neuromodulation are both high, and safety issue of this technique in clinical application remains a concern.

In this study, a much lower intensity ultrasound at the order lower than 10 mW/cm$^2$ is proposed to activate neurons via mechanosensitive ion channels in mammals' brain for potential clinical application. Mechanosensitve ion channels such as PIEZO and TRP channels and acid sensing ion channels (ASICs) (*Cheng et al., 2018*; *Murthy et al., 2017*; *Nilius and Honoré, 2012*) are considered the candidates likely responsive to ultrasound. Here, we aim to identify possible mechano-sensors in mouse brain that can respond to low-intensity ultrasound.

## Results

### Transcranial ultrasound-induced p-ERK in the cortex, hippocampus, and amygdala of mouse brain

We kept ultrasound exposure to below 10 mW/cm$^2$ ($I_{SATA}$) in our experiments to ensure safe therapeutic applicability. The phosphorylation of extracellular-signal-regulated kinase (p-ERK), an established indicator of immediate neuronal activation (*Gao and Ji, 2009*), was used to evaluate whether transcranial low-intensity ultrasound can stimulate neuronal activity in mouse brain. Mice with 1 min ultrasound exposure (*Figure 1A*) had shown significant increase of p-ERK positive cells in certain brain regions, such as the cortex (*Figure 1B–D*), hippocampus (*Figure 1E–G*), and amygdala (*Figure 1H–J*) as compared with those received sham treatment (*Supplementary file 1*). More specifically, increased p-ERK expression generally occurred upon ultrasound stimulation in the visual, somatosensory, auditory, temporal associations, retrosplenial, piriform, and entorhinal areas of mouse cortex (*Figure 1—figure supplements 1 and 2*). In hippocampus, CA1 and CA2 were dramatically lightened up with p-ERK signals in response to ultrasound whereas CA3 and dentate gyrus showed sparsely stimulated (*Figure 1—figure supplements 1 and 2*). In amygdala, the central amygdala nucleus showed the strongest p-ERK signals, while medial and basolateral also obviously increased in p-ERK signals (*Figure 1—figure supplements 1 and 2*). We also observed a consistently unchanged p-ERK staining in the paraventricular nucleus of hypothalamus (PVH) (*Figure 1—figure supplement 3*), revealing an intriguing regional specificity of the ultrasound response.

### Micropipette Guided Ultrasound as the Mechanical Stimuli with Combined Ultrasound and Acoustic Streaming

To determine whether low-intensity ultrasound can activate neurons mechanically, we next used in vitro calcium-imaging approach staining with the Oregon Green 488 BAPTA-1, AM to probe the possible ion channels responding to ultrasound mechanical stimulus in cultured cortical neurons. A micropipette was used to guide ultrasound to cultured cells (*Figure 2A*). The device can generate either an ultrasound induced pressure predominant condition (2000 mVpp, DF 0.05%, measured peak pressure versus the distance to the pipette tip shown in *Figure 2B*) or an acoustic streaming predominant condition (100 mVpp, DF 100%, flow pattern for the acoustic streaming shown in *Figure 2—figure supplement 1*); (*Chu et al., 2021*), or a mixed loading condition (700 mVpp, DF 20%) depending on duty cycle applied. Ultrasound pressure predominant conditions (up to 2000 mVpp, DF 0.05%) generated compressional stress on cells that were not able to elevate calcium responses (*Figure 2B* and *Video 1*), while acoustic streaming predominant conditions (up to 100 mVpp, DF 100%) invoked shear stress that could only activate little calcium responses in neurons (*Figure 2B*); whereas a mixed loading condition (700 mVpp, 20 % DF) effectively yielded much higher calcium responses (*Figure 2B* and *Video 2*). This response was also reproducible when we applied two additional live cell calcium staining reagents, including Fluo-4 (*Video 3*) and Fura-2 (*Figure 2—figure supplement 2A*, *Video 4*), and we observed similar responses upon micropipette guided ultrasound stimulations. To validate that the calcium response was not caused by a lipid bilayer damage, we repeated the stimulation on the same cell (*Figure 2—figure supplement 2B*), and found that despite an overwhelming photodecay problem due to continuous light exposure, there was some repeated calcium elevations at certain sub-cellular sites (white arrows, *Figure 2—figure supplement 2B*). The quantification of the neuronal calcium responses (*Figure 2—figure supplement 3A*) using Oregon Green 488 BAPTA-1 AM was presented in an average value (*Figure 2—figure supplement 3B*) and the experiments

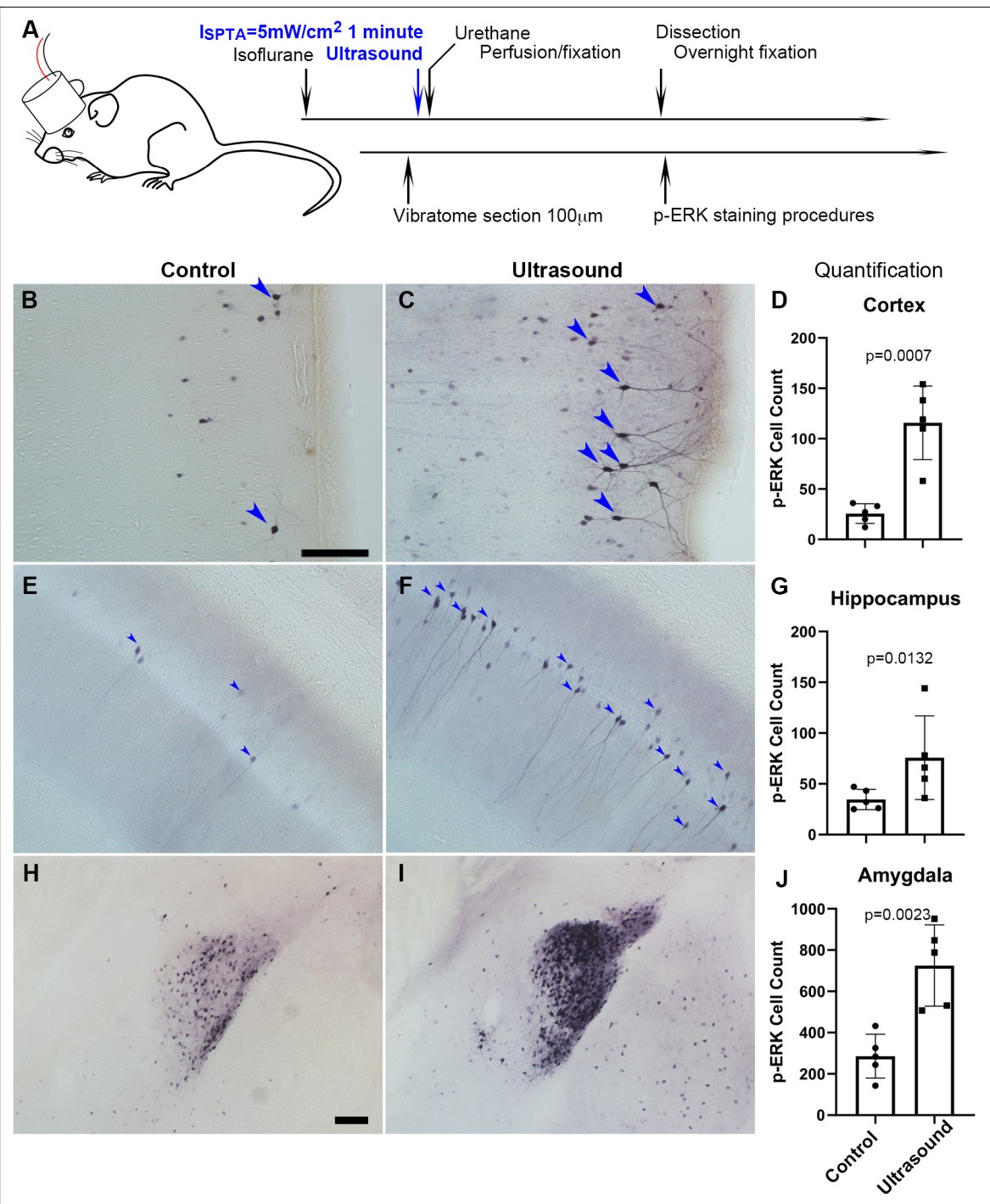

**Figure 1.** Transcranial ultrasound induces p-ERK expression in neurons of the cortex, hippocampus and amygdala of mouse brain. (**A**) Illustration depicting mouse head stimulated by 1 MHz transducer which was positioned in between the mouse nasal process of maxilla and the axis of mouse ear of an anesthetized mouse. (**B**) Micrograph representing cortical region with basal level of p-ERK staining in sham control mice (n = 5), scale bar 100 μm. Sham control mice were handled with similar procedures of placing transducer on the head without turning on the ultrasound function generator. (**C**) Micrograph representing cortical region with p-ERK staining stimulated by ultrasound ($I_{SPTA}$ = 5 mW/cm$^2$, 1 minute) (n = 5). (**D**) Quantitative bar graph of the number of p-ERK stained cells within comparable area of 1.224 mm$^2$ (Length 1275 μm, Width 960 μm), showing significant difference (*P* = 0.0007) of cell count by ImageJ. (**E**) Micrograph representing hippocampal region with basal level of p-ERK staining in sham control mice. (**F**) Micrograph representing hippocampal region with p-ERK staining in mice stimulated by ultrasound. (**G**) Bar graph showing quantification of significantly p-ERK different cell count (1.224 mm$^2$) (*P* = 0.0132) in hippocampal region. (**H**) Micrograph representing amygdala of sham controls. Scale bar 100 μm (**I**) Micrograph representing amygdala of ultrasound stimulated mice. (**J**) Quantification of amygdaloid significant difference (*P* = 0.0023) of p-ERK cell

*Figure 1 continued on next page*

*Figure 1 continued*

count (1.224 mm$^2$).

The online version of this article includes the following source data and figure supplement(s) for figure 1:

**Source data 1.** Source data for *Figure 1D, G, J*.

**Figure supplement 1.** Stitched images showing p-ERK expression of brain slice containing hippocampus, cortex and amygdala of sham control v.s ultrasound stimulated mice.

**Figure supplement 2.** Stitched images showing p-ERK expression of hippocampus and piriform cortex of sham control v.s ultrasound stimulated mice.

**Figure supplement 3.** The similar p-ERK expression of paraventricular nucleus of hypothalamus (PVH) of control untreated vs ultrasound stimulated mice.

were ended by a 0.01 % Saponin cell perforation to calibrate the maximum response (*Figure 2— figure supplement 3C*). This method was going to be utilized for the tests for inhibitors and dosage studies. To ensure that this method is reliable, cells were repeatedly stimulated and clear responses were observed even though the magnitude of responses typically dropped to 30–60% during the second stimulation (*Figure 2—figure supplement 3D*). The ultrasound predominant mode with only compressional stress cannot induce a response even when the stimulation was extended to 10 s (*Figure 2—figure supplement 4A*). Similarly, the prolong stimulation of acoustic streaming invoking shear stresses also only elevated mildly the amplitude of calcium response (*Figure 2—figure supplement 4B*). On the other hand, the combined compression stress with acoustic streaming can reproducibly elevated calcium response even in 1.5 s stimulation (*Figure 2B* and *Video 2*, *Figure 2—figure supplement 4C*). The ultrasound-induced calcium responses were dose-dependent with a threshold of 400 mVpp (8 kPa) and EC$_{50}$ of 700 mVpp (12 kPa) (*Figure 2C* and *Figure 2—figure supplement 4D-K*). The corresponding stress levels of ultrasound at 400 mVpp, 500 mVpp, 700 mVpp or 900 mVpp were 8 kPa, 8.72 kPa, 12 kPa, and 15.3 kPa, respectively.

## Neuronal calcium signal upon ultrasound simulation suppressed by ASIC channels inhibitors

The micropipette ultrasound mechanotransduction was pharmacologically tested with selective or non-selective blockers of mechanosensitive ion channels for Piezo, TRP, and ASICs. First, the treatment with Gadolinium (10 µM), a non-selective blocker of mechanically sensitive ion channels, partially suppressed the ultrasound-induced calcium signals (*Figure 2D*). Treating the cells with a selective Piezo inhibitor, GsMTx-4, led to a marginal (not significant) inhibition of the ultrasound-induced calcium elevation as compared with the vehicle control (*Figure 2E*). Instead, the treatment with a TRP blocker ruthenium red (1–10 µM) partially suppressed the calcium signals (*Figure 2F*), whereas the non-selective ASIC inhibitor, amiloride, totally abolished the calcium signals by micropipette ultrasound (*Figure 2G*). Above results suggested ASICs might be the major channels involved in micropipette ultrasound mechanotransduction. To narrow down the specific candidate of ASICs, ASIC1a inhibitor, PcTx1 (50 nM) was tested. PcTx1 (50 nM) significantly inhibited the calcium response by micropipette ultrasound (*Figure 2H* and *Figure 2—figure supplement 4C*). The relative inhibitions of above channel blockers were summarized in *Figure 2I*. We further tested the dose-dependent inhibition of PcTx1 on micropipette ultrasound and determined an IC50 of 0.2 nM, suggesting a homotrimeric ASIC1a is the mechanoreceptor in action (*Figure 3A* and *Figure 3—figure supplement 1 A-F*). To validate how ASIC1a activation would lead to calcium responses to ultrasound, we treated the cells with calcium chelating agent, ethylene glycol-bis(β-aminoethyl ether)-N,N,N',N'-tetraacetic acid (EGTA) (1–5 mM) to block the extracellular calcium. The results showed that calcium influx was absolutely essential (*Figure 3B* and *Figure 3—figure supplement 1G-H*). To examine whether endoplasmic reticulum calcium was involved in calcium signaling, we found the calcium surge of cells treated with the RyR inhibitor, JTV519 fumarate (10 µM) (*Figure 3—figure supplement 1G* and *I*) was partially inhibited, while as the IP3R inhibitor, (-)-Xestonspongin C (1 µM) was most inhibited (*Figure 3B*).

## ASIC1a mechano-response required cytoskeletal dynamics

Previous studies have suggested ASICs are involved in tether-mode mechanotransduction, which relies on intact cytoskeletal structures (*Cheng et al., 2018*; *Lin et al., 2009*). We likewise treated the cells with either actin polymerization inhibitor, cytochalasin D (5–10 µg/ml) or microtubule assembly

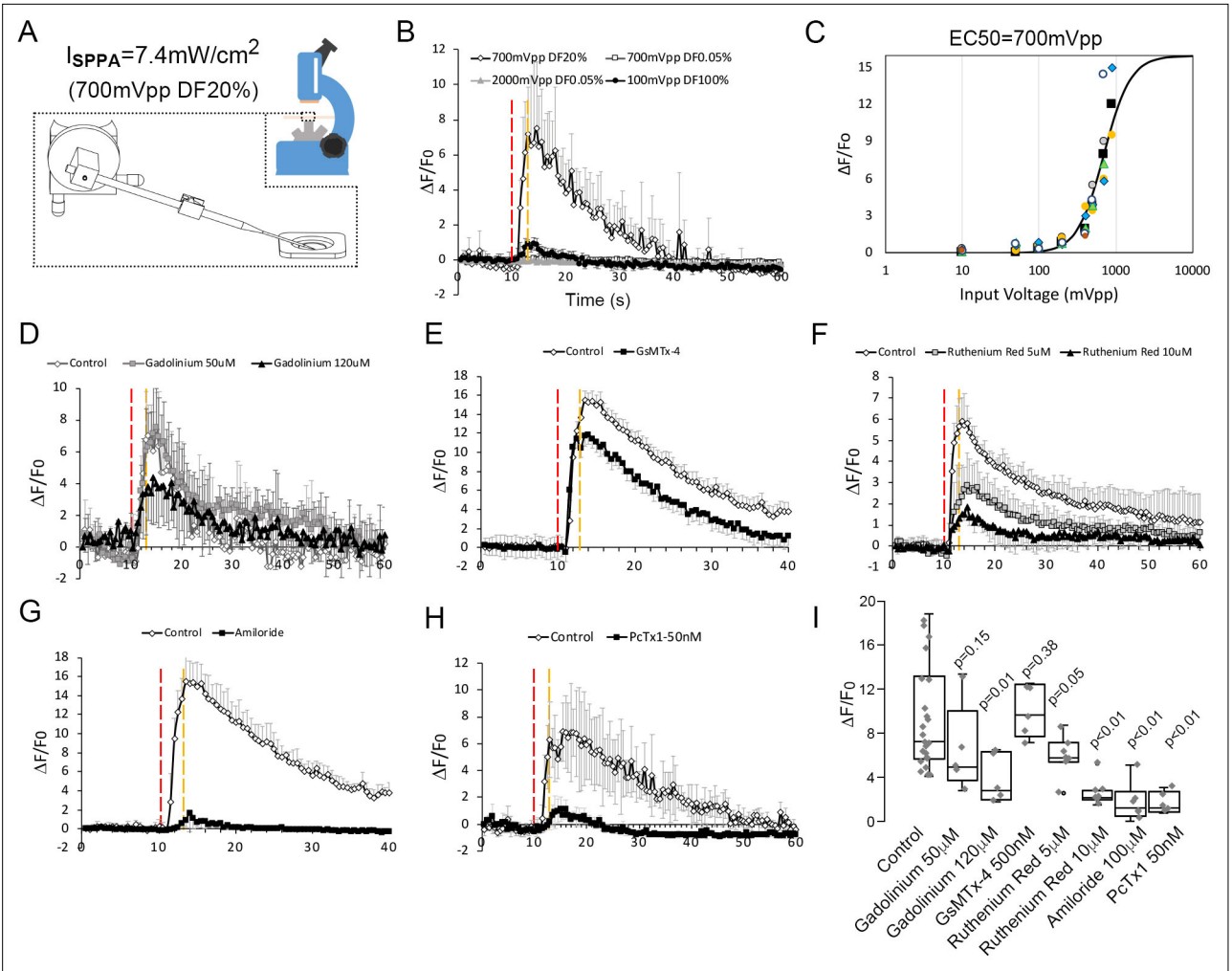

**Figure 2.** Neuronal calcium signals induced by micropipette-guided ultrasound suppressed by ASIC1a inhibitors. (**A**) A micropipette positioned to the cortical neurons cultured on a 30 mm cover slips mounted to a chamber coupled to microscope platform. Calcium signals recorded from neurons stained by Invitrogen Oregon Green 488 BAPTA-1, AM cell permeant. (**B**) Line graphs of averaged calcium signals in four neurons stimulated by micropipette ultrasound for 3 s with an input voltage 2000mVpp, duty factor (DF) 0.05 % (n = 5); or 100mVpp, DF100% (n = 3) or 700mVpp, DF20% (n = 4). The red-dotted line denotes start of the stimulation while the yellow-dotted line denotes the end. (**C**) Calcium responses as a function of micropipette ultrasound in 20%DF. Dose-dependent (input voltages from 10mVpp, to 900mVpp, DF20%) responses of ultrasound with an EC50 of 700mVpp is shown (n = 5). (**D**) Effects of gadolinium (120 µM), a non-selective blocker of mechanically sensitive ion channels, on calcium signals in cortical neurons (n = 5). (**E**) Effects of GsMTx-4 (500 nM), a selective Piezo inhibitor, on calcium signals in cortical neurons (n = 4). Control n = 4. (**F**) Effects of Ruthenium red (10 µM), a non-selective TRP inhibitor, on calcium signals in cortical neurons (n = 4). Control n = 6. (**G**) Effects of amiloride (100 µm), an ASICs family inhibitor, on calcium signals in cortical neurons (n = 4). (**H**) Effects of PcTx1 treated (50 nM), a selective ASIC1a inhibitor, on calcium signals in cortical neurons (n = 5). (**I**) Statistical analyses of channel blockers on ultrasound-induced calcium signals in cortical neurons. Control n = 21, Gadolinium (50 µM) n = 5, Gadolinium (100 µM) n = 5, GsMTx-4 (500 nM) n = 5, Ruthenium red (5 µM) n = 5, Ruthenium red (10 µM) n = 5, Amiloride (100 µM) n = 5, PcTx1 (50 nM) n = 5.

The online version of this article includes the following source data and figure supplement(s) for figure 2:

**Source data 1.** Source data for *Figure 2B and C*.

**Source data 2.** Source data for *Figure 2D*.

**Source data 3.** Source data for *Figure 2E and G*.

**Source data 4.** Source data for *Figure 2F*.

**Source data 5.** Sorce data for *Figure 2H*.

**Figure supplement 1.** Peak pressure stress and acoustic streaming pattern of micropipette guided ultrasound.

**Figure supplement 2.** Neuronal calcium response upon micropipette guided ultrasound stimulation.

**Figure supplement 3.** Quantification of neuronal calcium responses upon micropipette guided ultrasound stimulation.

*Figure 2 continued on next page*

*Figure 2 continued*

**Figure supplement 4.** Input voltage and duty factor settings for ultrasound dose and the neuronal calcium responses.

inhibitor, nocodazole (5–10 µg/ml). Indeed, inhibition of cytoskeletal dynamics could dose dependently and significantly suppress the calcium response stimulated by micropipette ultrasound (*Figure 3C–E*). The collated data revealed a novel mode of ultrasound mechanotransduction with a combination of compression force and shear force that activates ASIC1a channels in mouse neurons (*Figure 3F*).

## ASIC1a overexpression in CHO cells accelerated the calcium response upon ultrasound stimulation

We next distinguished ASIC1a's roles in causing the calcium response upon micropipette guided ultrasound stimulation in a heterologous expression system. We transfected *Asic1* cDNA (the plasmid was constructed using specifically *Asic1a* alternative spliced isoform) in the Chinese hamster ovary (CHO) cells that contain no endogenous ASIC1a. Cells treated with mock transfection reagent served as a control for comparison. Interestingly, CHO cells might contain some endogenous mechanosensitive machinery that can manifest a delayed calcium response to micropipette-guided ultrasound (*Figure 4A*, *Figure 4—figure supplement 1A*). In contrast, *Asic1*-transfected cells showed an immediate calcium response to the ultrasound (*Figure 4A–C*, *Video 5*, *Figure 4—figure supplement 1B*), indicating a role for ASIC1a in the ultrasound-mediated mechanotransduction (*Figure 4C*). The ultrasound-induced calcium responses were analyzed based on the area under curve (AUC) in different time points. Two-way ANOVA analysis showed both ASIC1a overexpression and ultrasound treatments significantly regulated the calcium response of CHO cells (*Supplementary file 1*) and the p value was 0.086 for the interaction of the two factors. The immediate calcium response of the *Asic1*-transfected CHO cells resembled that was detected in primary neurons (*Videos 2–4*). The calcium response calibrated by 0.01 % Saponin cell perforation could surge to the maximal Fura-2 ratio ($F_{340/380nm}$) (*Figure 4—figure supplement 1C, D*). In the context of 0.01 % Saponin, 2-way ANOVA analysis showed that ASIC1a overexpression did not contribute to the difference with an insignificant $p$pvalue ($F = 0.11$; $p = 0.74$) (*Supplementary file 1*) while cell perforation contributed significantly to the calcium responses ($F = 10.52$; $p < 0.0001$) (*Supplementary file 1*). Note that PcTx1 treatment did not affect the ultrasound-induced delayed calcium responses in vehicle transfected cells (*Figure 4E*), as ultrasound as a factor still contributed significantly ($F = 17.2$; $p < 0.0001$) to the changes of calcium while there was no significant ($F = 1.44$; $p = 0.15$) (*Supplementary file 1*) interaction of drug treatments with ultrasound stimulations (*Supplementary file 1*). In contrast, PcTx1 significantly inhibited ultrasound-induced calcium responses in *Asic1*-transfected cells (*Figure 4F*) ($F = 1.26$; $p = 0.29$) (*Supplementary file 1*), while

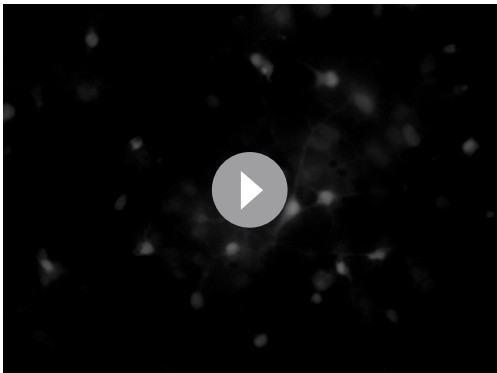

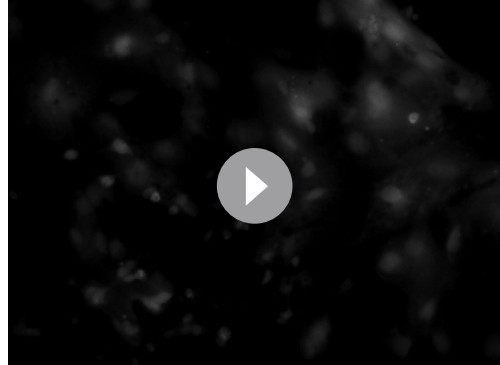

**Video 1.** Neuronal calcium signal cannot be induced by micropipette guided ultrasound with 2000mVpp input voltage and duty cycle 0.05 %. This setting induced produce predominantly ultrasound stimulation. When the dash line depicted micropipette tip appeared in the video, ultrasound function generator was turned on.

https://elifesciences.org/articles/61660/figures#video1

**Video 2.** Micropipette-guided ultrasound stimulation of neuronal calcium elevation. The setting was 700mVpp input voltage and duty cycle 20 % for 3 s. When the dash line depicted micropipette tip appeared in the video, ultrasound function generator was turned on. The setting generated both ultrasound and acoustic streaming effects.

https://elifesciences.org/articles/61660/figures#video2

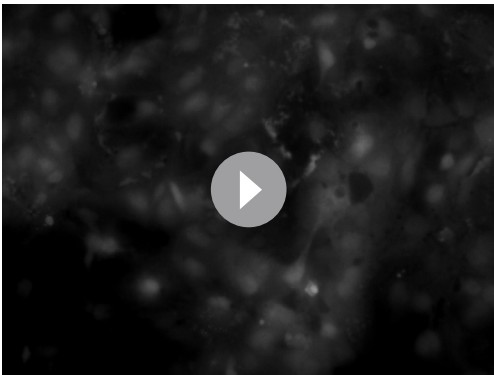

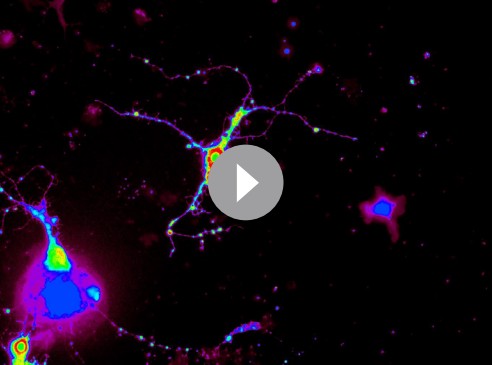

**Video 3.** Micropipette-guided ultrasound stimulation of neuronal calcium response. Ultrasound with 250mVpp input voltage and continuous waves stimulation. When the dash line depicted micropipette tip appeared in the video, ultrasound function generator was turned on. The setting generated both ultrasound and acoustic streaming effects.
https://elifesciences.org/articles/61660/figures#video3

**Video 4.** Micropipette-guided ultrasound with 400mVpp input voltage and duty factor 10 % induced the neuronal calcium signals captured by Fura-2 imaging methods. Calcium elevations were measured by fluorescence ratios of Fura-2 emission at wavelengths 340 nm/380 nm ($F_{340/380nm}$). Spectrum color coded fluorescence ratios in which the red color represents the highest ratio while purple color represents the lowest ratio of $F_{340/380nm}$. When the dash line depicted micropipette tip appeared in the video, ultrasound function generator was turned on. The ultrasound setting generated both ultrasound and acoustic streaming effects.
https://elifesciences.org/articles/61660/figures#video4

the two factors interact significantly ($F$ = 4.46, p < 0.0001) (*Supplementary file 1*). The overexpression of ASIC1a was validated by western analysis of CHO cell lysates (*Figure 4G*).

## Transcranial ultrasound treatments promoted neurogenesis in dentate gyrus

We next investigated whether the low-intensity ultrasound stimulation in mouse brain could lead to a favorable outcome in terms of adult neurogenesis. We selected doublecortin (DCX) as a marker for neurogenesis in dentate gyrus (*Germain et al., 2013*; *Jin et al., 2010*; *Rao and Shetty, 2004*). Compared to the non-treated controls, after three consecutive days of 5 min ultrasound treatments (5 mW/cm$^2$), DCX staining in dentate gyrus at day 4 and day seven showed a significant twofold increase (*Figure 5*). The results indicated that repeated stimulation of low-intensity ultrasound on mouse brain might achieve beneficial neural modulation and lead to neurogenesis in dentate gyrus.

## Neurogenesis marked by DCX induced by ultrasound was partially compromised by *Asic1* knockout (specifically designed for *Asic1a* alternative spliced isoform)

We performed the same experiments on either the wildtype mice or the *Asic1* knockout (*Asic1*$^{-/-}$) mice for 3 consecutive days of 1 min ultrasound treatments (5 mW/cm$^2$) and sacrificed the mice at day seven to quantify the ultrasound effects on DCX staining in these mice. We observed a reproducibly significant increase of DCX staining in the ultrasound treated group while quantification and student t-test analysis indicated that the increase was partially compromised in *Asic1*$^{-/-}$ (*Figure 6A–I*). We tested whether there was an interaction between the two factors, that is the two-way ANOVA analysis results showed that the ultrasound treatment ($F$ = 9.4; p = 0.0098) and the *Asic1*$^{-/-}$ ($F$ = 26.35; p = 0.0002) in regulating the DCX-positive cell counts (*Supplementary file 1*). In addition, there was no significant interaction ($F$ = 0.22; p = 0.65) of these two factors while both factors contributed significantly and independently to the DCX cell counts (*Supplementary file 1*).

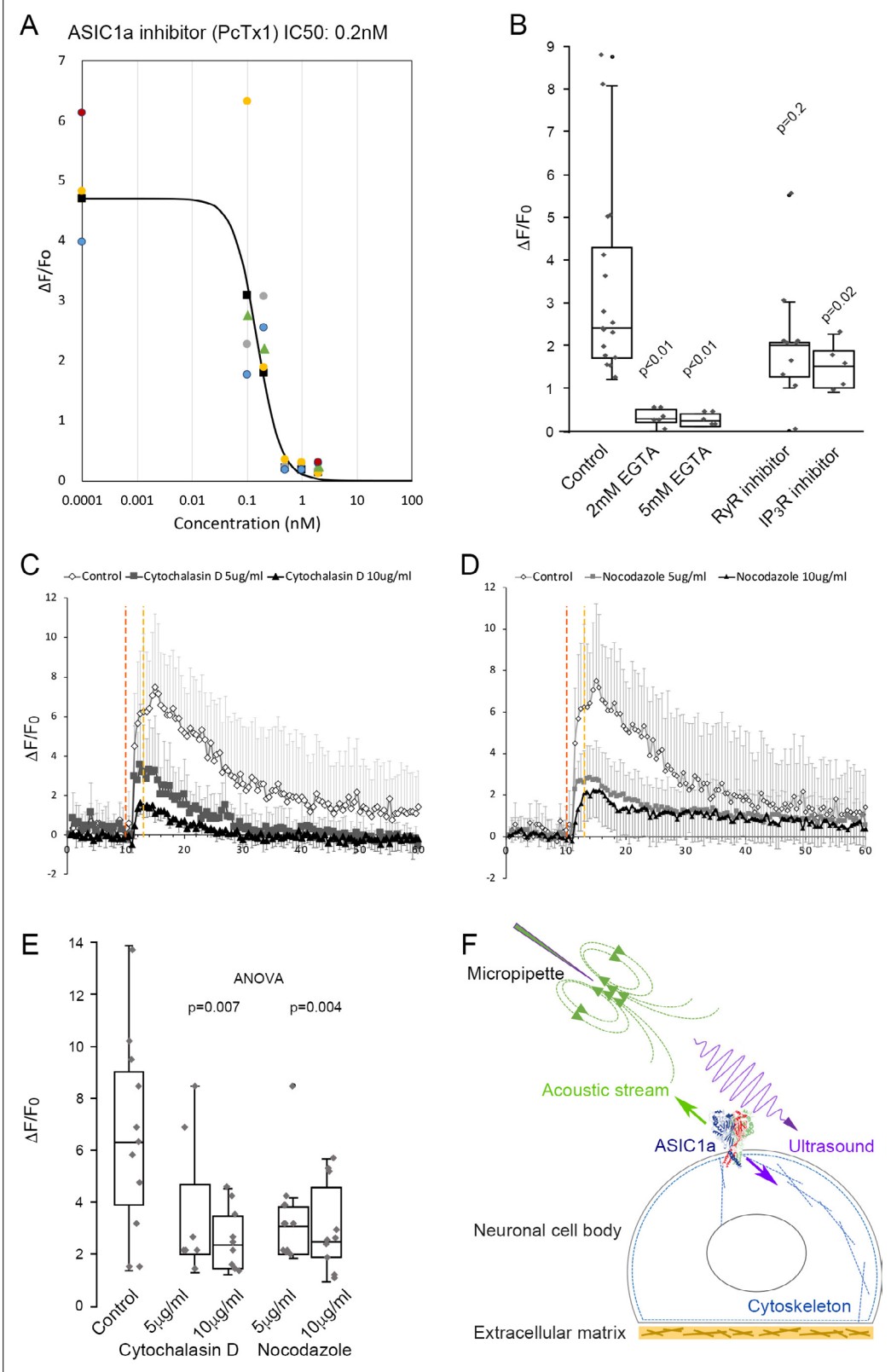

**Figure 3.** ASIC1a as a mechanoreceptor responsive to mechanical stimuli with combined ultrasound and acoustic streaming. (**A**) PcTx1 dose-dependent inhibition curve of calcium responses induced by micropipette ultrasound of 700mVpp, DF20% for 3 s (n = 5). (**B**) Whisker plots showing comparison of peak $\Delta F/F_0$ within 3–5 s upon micropipette ultrasound (700mVpp, DF20%, 3 s) stimulation in the untreated control primary neurons (n =

*Figure 3 continued on next page*

*Figure 3 continued*

16), 2- or 5 mM EGTA-treated neurons (n = 6 or 4, respectively), RyR inhibitor JTV519 fumarate (10 µM) treated neurons (n = 10), or $IP_3R$ inhibitor (-)-Xestonspongin C (1 µM)-treated neurons (n = 5). Student t-test with p value compared to control listed above the whisker plot. (**C**) Graph showing calcium response of actin polymerization inhibitor Cytochalasin D (5–10 µg/ml)-treated neurons (n = 7 and n = 9, respectively) compared to untreated control (n = 5). (**D**) Calcium signals showing the effect of the microtubule assembly inhibitor nocodazole (5–10 µg/ml) on neurons (n = 8 and n = 10, respectively). (**E**) Whisker plots showing comparison of peak $\Delta F/F_0$ within 3–5 s upon micropipette ultrasound (700mVpp, DF20%, 3 s) stimulation in the untreated control primary neurons (n = 11), cytochalasin D 5- or 10 µg/ml treated neurons (n = 7 or n = 9, respectively), nocodazole 5- or 10 µg/ml treated neurons (n = 8 or n = 10, respectively). Statistical p values of one-way ANOVA analysis were listed to show the significance of treatment. (**F**) Cartoon depicting ultrasound stimulating ASIC1a in the cell body of a neuron under the micropipette ultrasound stimulation. Green arrow represents the pulling force of acoustic stream and purple arrow represents the compression force of ultrasound that results in cytoskeletal rearrangement.

The online version of this article includes the following source data and figure supplement(s) for figure 3:

**Source data 1.** Source data for *Figure 3A*.

**Source data 2.** Source data for *Figure 3B*.

**Source data 3.** Source data for figure *Figure 3C-E*.

**Source data 4.** Source data for *Figure 3E, G*.

**Figure supplement 1.** **A–F**The dose-dependent calcium response of PcTx1, an ASIC1a inhabitation, under micropipette stimulation (700mVpp, 20%DF, check).

## *Asic1*[-/-] suppressed transcranial ultrasound induced P-ERK in mouse brain

To study whether ASIC1a is also responsible for the responses of p-ERK in mouse brain, we employed *Asic1*[-/-] and *Asic3*[-/-] in the vibratome brain slices p-ERK staining experiments. The inclusion of *Asic3*[-/-] is to elucidate the role of peripheral nerves in brain activation, as ASIC3 is highly expressed in in somatosensory neurons, trigeminal ganglion neurons, and spiral ganglion neurons. Comparing to the mock controls (*Figure 7A, H and O*), the p-ERK cell counts in wildtype mice upon transcranial ultrasound stimulations (*Figure 7B, I and P*) were significantly increased (*Figure 7*). Comparing to the mock controls (*Figure 7C, J and Q*), the cell counts increase of p-ERK staining in *Asic1*[-/-] was partially decreased in hippocampal region while greatly reduced in cortical and amygdala regions (*Figure 7D, K and R*). The reduction of p-ERK cell counts caused the difference between mock control and ultrasound stimulation to be statistically insignificant in *Asic1*[-/-] mice (*Figure 7G, N and U*). On the other hand, *Asic3*[-/-] mice showed a more consistent lower background of p-ERK in mock controls (*Figure 7E, L and S*) and exhibited a significant increase of p-ERK cell counts (*Figure 7F, M and T*) to the ultrasound treatment. Quantification of p-ERK responses in these three genotypes of mice led us to conclude that ASIC1a plays an important role in mediating transcranial ultrasound stimulation in mouse brain. The two-way ANOVA analysis of p-ERK cell counts showed that there was only an interaction of two factors, namely genotype and ultrasound treatment in the cortex (*F* = 6.45, p = 0.0037) but not in hippocampus and amygdala (*Supplementary file 1*). To test whether there is a role of peripheral nerves in mediating ultrasound stimulation, we included the *Asic3*[-/-] in our p-ERK response phenotypes studies and indeed the genotype did not reduce the activation of p-ERK as *Asic1*[-/-] did. These results indicated that the p-ERK response in mouse brain is likely directly caused by the transcranial ultrasound instead of caused by the secondary effects due to the neurons wired with auditory circuits or other sensory circuits, as ASIC3 being mainly expressed in somatosensory neurons and spiral ganglion neurons (*Lin et al., 2016*; *Wu et al., 2009*).

To further address the cell types showing p-ERK response upon ultrasound stimulation, we performed immunofluorescent co-staining of several markers, such as NeuN, NMDAR, GAD67, and PV. There was an obvious p-ERK co-staining with NeuN (94 % or 197/209) however these cells were not NMDAR positive (*Figure 7—figure supplement 1*). On the other hand, there was a very small population co-staining with interneuron marker GAD67 (4.5 % or 10/223) and even smaller population co-staining PV (0.9 % or 2/211). An ASIC1a-specific antibody for IHC or IF staining will shed light on whether the p-ERK-responsive cells are ASIC1a-positive neurons. Nevertheless, further study is

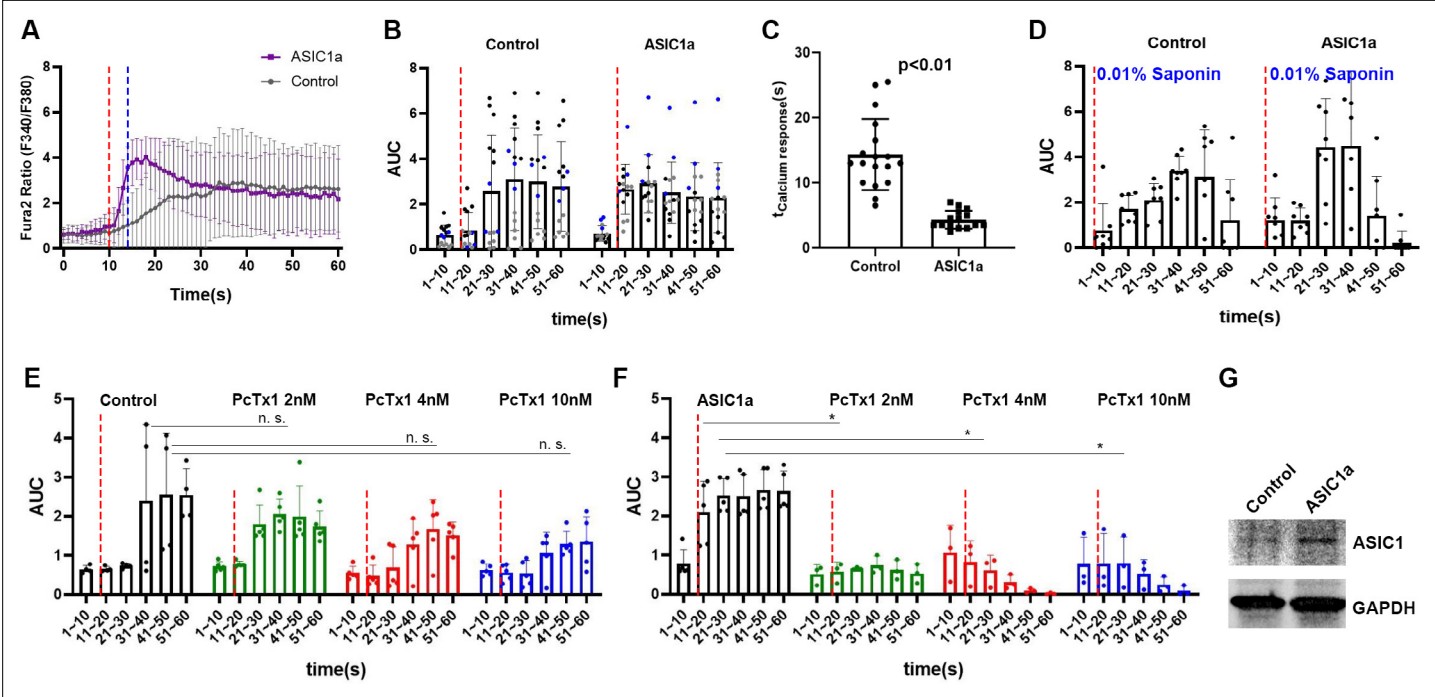

**Figure 4.** ASIC1a overexpression showed a fast calcium response upon micropipette-guided ultrasound stimulation in CHO cells. (**A**) Invitrogen Fura-2, AM, cell permeant (Fura-2) stained CHO cells. Fluorescence ratios of Fura-2 emission at wavelengths 340 nm/380 nm ($F_{340/380nm}$) were recorded and averaged line graphs were shown. $F_{340/380nm}$ ratio values plotted against time were shown in *Figure 4—figure supplement 1A*, B. Ultrasound stimulation is indicated by the red dashed lines at time point 10 s for a duration of 3 s. The blue dashed line indicates the time of ultrasound termination. Control n = 18, ASIC1a overexpressed n = 15. (**B**) Area under curve (AUC) of $F_{340/380nm}$ were plotted in a 10 s bin manner. Each dot represents a single cell quantified. Three batches of experiments were represented by three different colors. Refer to *Supplementary file 1* for the two-way ANOVA analysis of this graph. Control n = 14, ASIC1a overexpressed n = 15. (**C**) Calcium response time determined by the maximum $F_{340/380nm}$ was significantly shortened by ultrasound stimulation compared to the sham transfected controls. Control n = 18, ASIC1a overexpressed n = 15. (**D**) Cell perforation treatment with 0.1 % saponin in HHBS after the experiments for internal calibration of maximal response. Refer to *Supplementary file 1* for the two-way ANOVA analysis of this graph. Control n = 7, ASIC1a overexpressed n = 7. (**E**) CHO cells calcium response either with or without PcTx1 treatments. Refer to *Supplementary file 1* for the two-way ANOVA analysis of this graph. Control n = 4, PcTx1 (2 nM) n = 5, (4 nM) n = 5, (10 nM) n = 5. (**F**) ASIC1a overexpressing CHO cells either with or without PcTx1 treatments. Refer to *Supplementary file 1* for the two-way ANOVA analysis of this graph. Control n = 5, PcTx1 (2 nM) n = 3, (4 nM) n = 3, (10 nM) n = 3. (**G**) Western analysis of ASIC1a comparing the sham transfected control and *Asic1*-transfected cells. GAPDH detection serves as an internal control.

The online version of this article includes the following figure supplement(s) for figure 4:

**Source data 1.** Source data for *Figure 4A*.

**Source data 2.** Source data for *Figure 4B and C*.

**Source data 3.** Source data for *Figure 4D*.

**Source data 4.** Source data for *Figure 4E*.

**Source data 5.** Source data for *Figure 4F*.

**Figure supplement 1.** Effect of ASIC1a over expression in CHO cells on calcium response upon micropipette guided ultrasound stimulation.

needed to have a comprehensive picture of the cell types that are specific to ultrasound-activated response.

## Discussion

Accumulating evidence has shown ASICs are involved in different types of mechanotransduction in the sensory nervous system, including nociception, baroreception, proprioception, and hearing (*Cheng et al., 2018*; *Lin et al., 2016*; *Chen and Wong, 2013*). However, the mechanosensitive role of ASICs in the brain is still not known, although ASIC1a is a predominant acid sensor modulating neural activity in physiological and pathological conditions (*Baron and Lingueglia, 2015*; *Wemmie et al., 2013*).

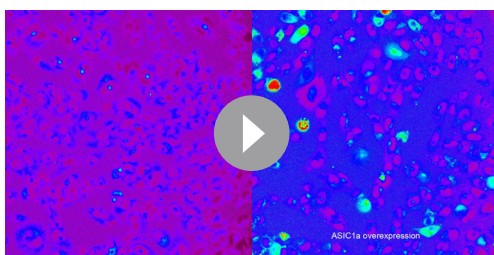

**Video 5.** CHO cells calcium response induced by micropipette guided ultrasound. Right panel was CHO cells overexpressing ASIC1a while left panel was showing transfection sham control cells. Calcium elevations were measured by fluorescence ratios of Fura-2 emission at wavelengths 340 nm/380 nm ($F_{340/380nm}$). Spectrum color coded fluorescence ratios in which the red color represents the highest ratio while purple color represents the lowest ratio of $F_{340/380nm}$. When the dash line depicted micropipette tip appeared in the video, ultrasound function generator was turned on. The ultrasound setting was 600mVpp input voltage and duty cycle 10 % for 3 s. This setting generated both ultrasound and acoustic streaming effects.

https://elifesciences.org/articles/61660/figures#video5

Here, we demonstrated low-intensity ultrasound could modulate neural activity in mouse brain and directly activate neurons (*Figure 1*) via an ASIC1a-depenent manner (*Figures 2–3*). While the current view of transcranial ultrasound activation of neurons in brain is through the auditory nerves (*Sato et al., 2018*), our results from *Asic3⁻/⁻* mice suggest that the peripheral nerves may not play a role in the activation of p-ERK in mouse brain by low intensity ultrasound. Alternatively, the low-intensity ultrasound-mediated mechanotransduction may act via a channel subtype-dependent manner specific for ASIC1a but not for other ASIC subtypes as shown in dextrose prolotherapy (*Han et al., 2021*). Moreover, repeated transcranial low-intensity ultrasound stimulations are safe and able to elicit adult neurogenesis in mouse brains (*Figure 4*).

Although ASIC1a was determined as the molecular determinant involved in low-intensity ultrasound mechanotransduction, non-selective mechanosensor inhibitors such as gadolinium and ruthenium red were partially suppressing the calcium response triggered by micropipette ultrasound (*Figure 2G and H*). Of note gadolinium also blocked ASICs in µM to mM ranges. However, we cannot rule out a role of TRP channels in the micropipette ultrasound mechanotransduction, because there is no evidence showing ruthenium red can also inhibit ASICs. Previous studies have proposed a role of TRP for ultrasound-mediated mechanotransduction while high-intensity ultrasound was applied. More studies are required to validate the role of TRP in low-intensity ultrasound mechanotransduction or the unexpected role of ruthenium red in neuronal ASIC1a signaling pathways.

ASIC1a is widely expressed in the brain and could form as homotrimeric and heterotrimeric channels with different sensitivity to PcTx1 inhibition (*Joeres et al., 2016*; *Sherwood et al., 2011*). Specifically, heterotrimeric ASIC2b/ASIC1a can be inhibited 50 % by approximately 3 nM (*Sherwood et al., 2011*) PcTx1 and ASIC1a/ASIC2a heterotrimeric can be inhibited by 50 nM (*Joeres et al., 2016*), whereas 0.5 nM to 1 nM can inhibit the homotrimeric ASIC1a (*Escoubas et al., 2003*; *Saez et al., 2011*). Therefore, since PcTx1 in low-doses effectively inhibited ASIC1a-mediated calcium signal by micropipette ultrasound, homotrimeric ASIC1a channels may be the predominant subtype involved in ultrasound mechanotransduction in cortical neurons (*Figure 3A* and *Figure 3—figure supplement 1*).

To explain the mode of ultrasound induced ASIC1a mechanotransduction, we hypothesized a physical effect of micropipette ultrasound at cell level; which the acoustic streaming imposes shear stress on cell apical surfaces while ultrasound exerts compressional stresses throughout the cells. Considering the combinatorial forces mode in vitro, we argue, in a mixed loading condition (*Figures 2B and 3 F*), the extracellular domains of ASIC1a are under shear force pulling the protein to the flow direction while the intracellular domains of the ASIC1a are connected to cortical actin or other cytoskeleton, which experiences dynamic reorganization coupling with membrane withdrawals in response to ultrasound (*Chu et al., 2019*; *Lim et al., 2020*). As such, mechano-signal triggered ASIC1a, essentially a sodium channel, results in the intracellular calcium elevation possibly by activating voltage-gated calcium channels (*Boillat et al., 2014*).

The condition in vivo on the other hand (*Figure 3—figure supplement 1*), is accomplished differently. Neurons are embedded in extracellular matrixes (earthy yellow color) such as laminin, poly-lysine, or poly-ornithine in the brain. ASIC1a is N-glycosylated at N366 and N393, both residues extracellular located (*Jing et al., 2012*). While N-glycosylation is reported to be involved in the surface trafficking and dendritic spine trafficking of ASIC1 (*Jing et al., 2012*; *Kadurin et al., 2008*), the N-glycosylation

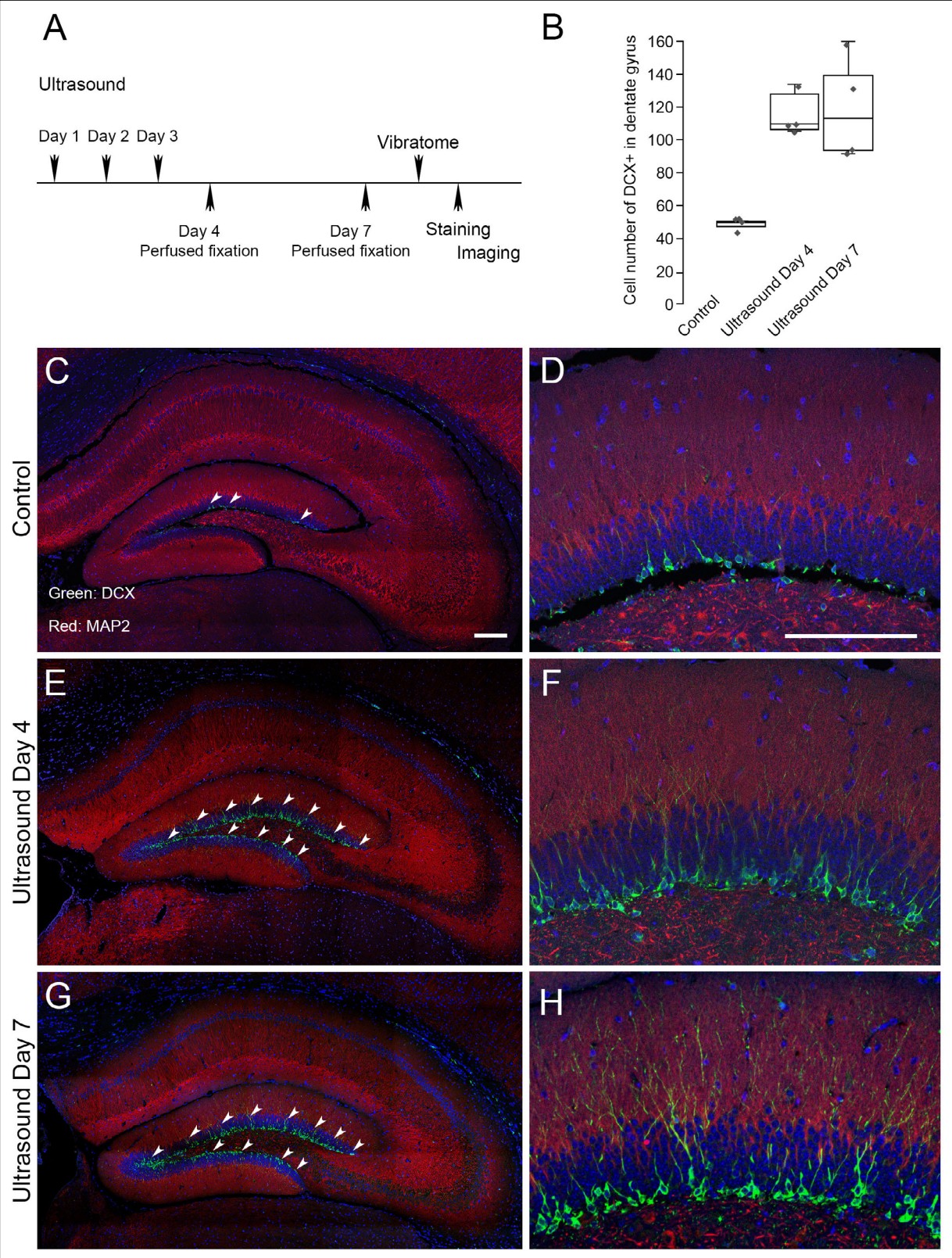

**Figure 5.** Neurogenesis in dentate gyrus induced by repeated transcranial ultrasound treatments. (**A**) The mice were treated three consecutive days by ultrasound of 4 mW/cm$^2$, 1 % for 5 min, subsequently perfused fixed at day 4 or day 7 and brains were dissected from the head and sectioned for immunofluorescence procedures. The DCX staining in dentate gyrus of treated mice were compared to control untreated one. (**B**) Cell count with clear DAPI stained nucleus surrounded by DCX markers compared in control, day 4 and day 7 post-ultrasound treatments. Statistical analysis: p = 0.0013 and

*Figure 5 continued on next page*

*Figure 5 continued*

F ratio = 15.18 in one-way ANOVA (n = 4). (**C, D**) Representative micrograph of untreated mice. The vibratome coronal brain sections (100 µm) of dentate gyrus region immunofluorescently stained for DCX (green) and MAP2 (red). Blue color indicates DAPI stained nuclei. Representative micrograph showing. (**E, F**) Representative micrograph from ultrasound treated mice fixed at day 4. (**G, H**) Representative micrograph from ultrasound treated mice fixed at day 7. Scale bar 200 µm.

The online version of this article includes the following figure supplement(s) for figure 5:

**Source data 1.** Source data for *Figure 5B*.

of many proteins has been known to be important for adhesion and migration (*Gu, 2012*; *Medina-Cano et al., 2018*; *Stevens and Spang, 2017*), implicating the extracellular matrix interacting nature of N-glycans. When ultrasound is applied to the brain, the acoustic pressure exerted through extracellular matrixes, can possibly activate ASIC1a via a cytoskeletal-dependent manner (*Figure 3—figure*

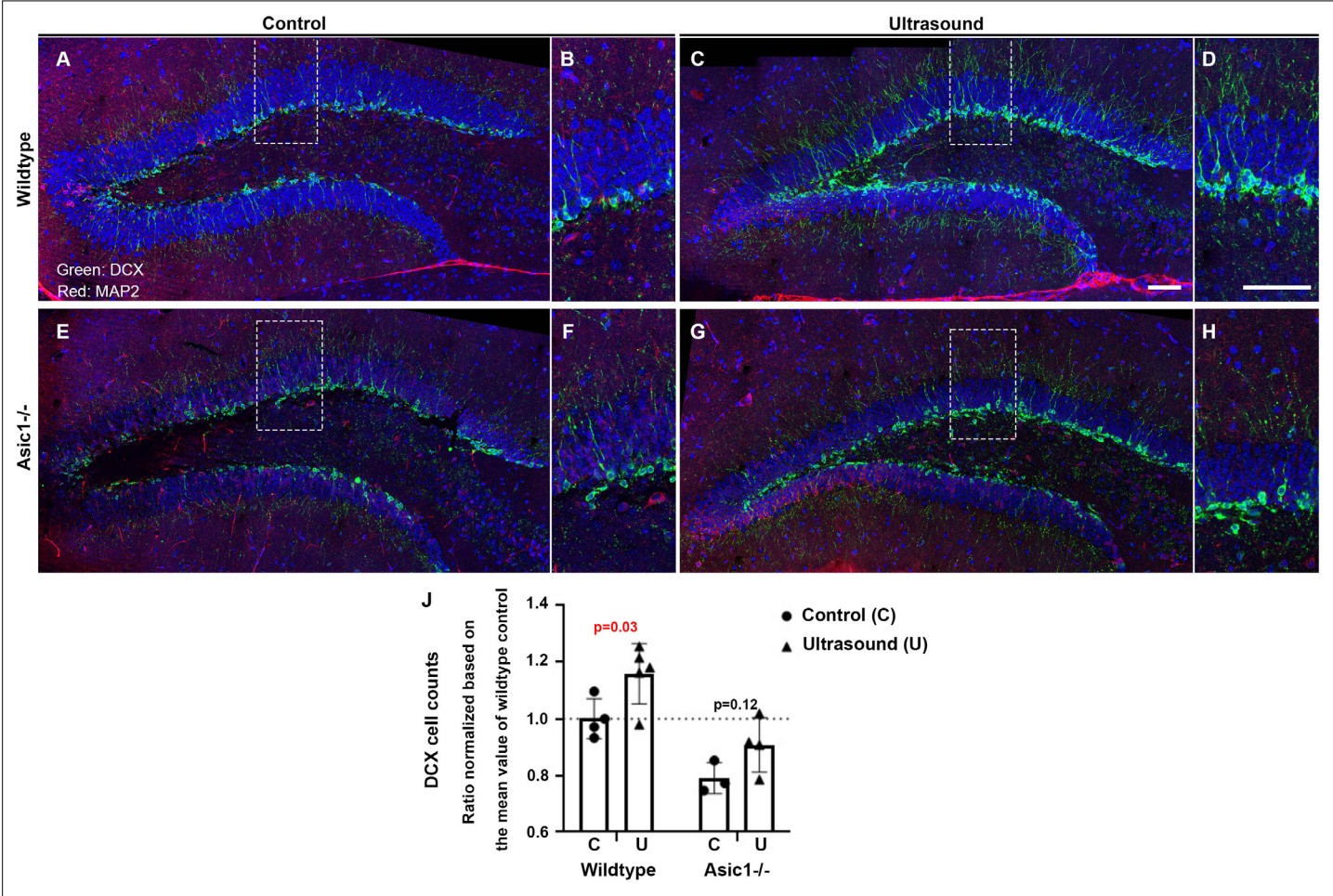

**Figure 6.** DCX staining-positive cells are increased but partially compromised by *Asic1*−/− after consecutive 3 days of ultrasound treatments. (**A**) Micrographs stitched to show the representative DCX staining pattern (green fluorescence) of the dentate gyrus (DG) in the 5 weeks old mice of wildtype sham treated controls. (**B**) Magnified DCX positive cells in wildtype control DG. (**C, D**) DCX-positive cells increased significantly upon three continuous days of ultrasound treatments. (**E, F**) Representative stitched micrographs of sham treated *Asic1*−/− dentate gyrus. (**G, H**) The increase of DCX staining upon ultrasound stimulation partially compromised by *Asic1*−/−. (**I**) Quantitative analysis of DCX cell counts/mm in the 100 µm brain slices with clear DCX and DAPI staining using confocal microscopy scanning stacks of 8–10 z-planes. Scale bar 100 µm. There were two batches of mice of 5 weeks and 7 weeks old and the cells counts from all the z-stacks were normalized based on the mean value of wildtype controls to include both batches of mice. Each data point represents quantification of one animal; wildtype control n = 4, wildtype ultrasound treated n = 5, *Asic1*−/− control n = 3, *Asic1*−/− ultrasound stimulated n = 4. Refer to *Supplementary file 1* for the two-way ANOVA analysis of this graph.

The online version of this article includes the following source data for figure 6:

**Source data 1.** Source data for *Figure 6J*.

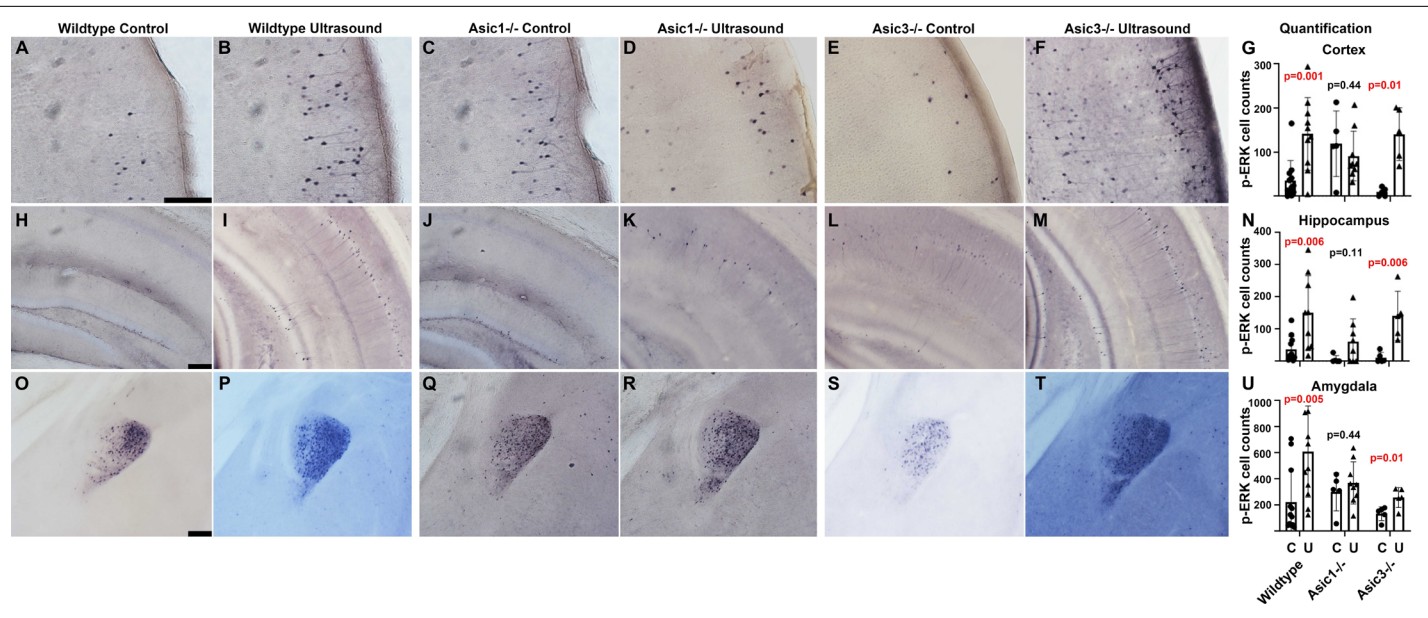

**Figure 7.** *Asic1-/-* suppressed the p-ERK cell count increases in cortex, hippocampus, and amygdala of mouse brain. The IHC stained brain slices of wildtype mice, *Asic1-/-* mice and *Asic3-/-* mice. Mice of all genotypes were randomly assigned to sham treatment group and ultrasound treatment group. The quantification of p-ERK-positive cells were performed using ImageJ with setting of threshold and particle sizes that representing the actual staining pattern. (**A–F**) Micrographs depicting p-ERK IHC staining in the cortex of the vibratome brain slices. (**G**) Quantification comparing cortical p-ERK-positive cells in three different genotypes of mice either mock treated or ultrasound stimulated. (**H–M**) IHC micrograph depicting p-ERK staining in hippocampus. (**N**) Quantification comparing hippocampal p-ERK-positive cells in mice with indicated the genotypes and treatments. (**O–T**) IHC micrographs depicting p-ERK staining in amygdala. (**U**) Quantification comparing p-ERK-positive cells in amygdala. Scale bar 100 μm. Each data point represents the total cell count of one mouse brain; wildtype control n = 11, wildtype ultrasound treated n = 10, *Asic1-/-* control n = 5, *Asic1-/-* ultrasound stimulated n = 9, *Asic3-/-* control n = 5, *Asic3-/-* ultrasound stimulated n = 5. Refer to **Supplementary file 1** for the two-way ANOVA analysis of this graph.

The online version of this article includes the following source data and figure supplement(s) for figure 7:

**Source data 1.** Source data for *Figure 7G, N and U*.

**Figure supplement 1.** Co-staining of p-ERK with neuronal markers such as NeuN, NMDAR, GAD67, and PV (Green fluorescence).

supplement 1*), in addition to other mechanosensitive machineries such as PIEZO and TRPV4 (*Poole et al., 2014*; *Servin-Vences et al., 2017*), since these mechanoreceptors have all been proven to be triggered by indentation of substrates. Consequently, this leads to an activation of the cells manifesting as ERK phosphorylation.

Since DCX has been accepted as a surrogate mark for neurogenesis in dentate gyrus (*Jin et al., 2010*; *Salvi et al., 2016*), increased DCX-positive cells in dentate gyrus with a consecutive 3 day of ultrasound treatment suggests a therapeutic implication. Of note, DCX plays multiple roles in brain development including hippocampal pyramidal neuronal lamination, cortical neuronal migration, and axonal wiring (*Germain et al., 2013*; *Koizumi et al., 2006*). The use of ultrasound in developing brains should be extremely cautious.

In conclusion, here we provide evidence that a clinically safely low-intensity transcranial ultrasound could modulate neuronal activity in mouse brain. The low-intensity ultrasound can directly activate neurons via ASIC1a, which provides a molecular basis for future development of ultrasound neuromodulation.

## Materials and methods
### Ultrasound devices and stimulation parameters

Two different ultrasonic setups were used in our study. We used a commercial 1MHz transducer (C539-SM, Olympus, Tokyo, Japan) for mouse brain stimulation in the in vivo experiment (*Figure 1A*). Simulation for calcium imaging was done with micropipette ultrasound (*Chu et al., 2021*) attaching a 1MHz transducer (15 mm in diameter). Schematics of experimental setup for

ultrasound calcium imaging is shown in *Figure 2A*. All the transducers were controlled by a function generator (Tektronix AFG1022, Beaverton, OR, USA) through a power amplifier (E&I 210 L, Rochester, NY, USA). The input voltage was 900 mVpp with a duty factor of 1 % at 1 kHz pulse rate for the in vivo stimulation. We characterized cellular exposure to ultrasound using a hydrophone (HGL-1000, Onda, Sunnyvale, CA, US) immersed in water. The intensity used for in vivo animal experiments was 5 mW/cm$^2$ (I$_{SPTA}$), and 7.4 mW/cm$^2$ (I$_{SPPA}$, at 700 mVpp) for micropipette in vitro experiment. These intensity values are within the range of not causing any side effects from ultrasound, like heat and cavitation.

Exploring upstream mechanoreceptors requires a calcium imaging assay that can respond to ultrasound stimulation repeatedly and reliably so that the effect of inhibitors can be demonstrated clearly. Micropipette ultrasound offers a wide range controllability. To select appropriate parameters for calcium experiments, we tested two extreme conditions: one with high input voltage and low duty factor (1500 mVpp, 0.05 % duty factor) for a predominant ultrasound stimulus and the other with low input voltage (100 mVpp) and continuous waves for a predominant acoustic streaming stimulus. As a predominant ultrasound stimulus, micropipette ultrasound exhibits a point source characteristic (*Figure 2—figure supplement 1A*). As a predominant streaming stimulus, micropipette ultrasound yields an inward flow pattern (*Figure 2—figure supplement 1B*).

The position of the micropipette is adjusted to stimulate the neurons as shown in (*Figure 2—figure supplement 1C*). The distance from the tip of micropipette to the apical membrane of the cells is approximately controlled to be 20 μm (*Figure 2—figure supplement 1D*).

## Animals

All animal procedures complied with the guidelines of the Institutional Animal Care and Use Committee in Academia Sinica, Taipei, Taiwan. *Asic1−/−* mice were a gift from Dr. CC Lien of NYCU of Taiwan and generated by crossing *Asic1* conditional KO (*Asic1$^{f/f}$*) mice (*Wu, 2013*) with protamine-Cre mice (*Lin et al., 2015*). *Asic3*-knockout/*eGFP-f*-knockin mice (*Asic3$^{-/-}$*) were generated based on the *Accn3* gene; in brief the design was mainly the targeting allele with a 6 kb long arm (*Hinc*II~*Hinc*II DNA fragment located 82 bp downstream the transcription start site) and a 240 bp short arm upstream of the ATG translation start site of *Accn3* was used for homologous recombination (*Lin et al., 2016*). Either wildtype, or *Asic1$^{-/-}$* or *Asic3$^{-/-}$* C57B6/J mice of 6–8 weeks were shaved under isoflurane anesthesia the day previous to ultrasound stimulation. The mice were randomly assigned to be either sham treated by placing ultrasound transducer on top of head or really exposed to ultrasound for 1 min to evaluate neuronal activities in the mouse brain after the ultrasound stimulation under isoflurane anesthesia. Immediately after the treatment, mice were first anesthetized with urethane (1.5 g/kg; intraperitoneal) and perfused transcardially with 25 ml 0.02 M Tris buffer saline (1 x TBS, pH7.4, at 4 °C) and then 25 ml cold fixative (4%[w/v] formaldehyde, 0.02 M TBS (pH7.4, at 4 °C)).

## Brain histology and immunohistochemistry

Mouse brain was dissected and post-fixed with 4 % formaldehyde at 4 °C for 16 hr; tissues were sectioned with Vibratome 1000 Plus (Rankin Biomedical, Holly, MI) at 100 μm thickness and incubated with antibody in free-floating method. For ABC-DAB-Nickel staining, tissue sections were first bleached in 1 x TBS containing 0.03 % H$_2$O$_2$ for 30 min, and then blocked in TBST (TBS +0.05 % Triton X-100) containing 5 % bovine serum albumin (BSA) (Sigma-Aldrich, St. Louis, MO, USA) and 5 % normal goat serum (NGS from Jackson ImmunoResearch Laboratories, West Grove, PA, USA) at room temperature for 60 min, and incubated with Rabbit polyclonal Phospho-p44/42 MAPK (ERK1/2) (Thr202/Tyr204) primary antibody [(1:500) #9101, Cell Signaling Technology, Danvers, MA, USA] diluted in blocking solution overnight at 4 °C. Sections were then washed three times with TBST and incubated with secondary biotinylated goat-anti-rabbit antibodies (1:1000, Vector Laboratories, Burlingame, CA, USA) for 1 hr at room temperature. After three TBST washes, sections were incubated in the Avidin-Biotin pre-mix solution (1:200, Vector Laboratories, Burlingame, CA, USA). After 3 1xTBS washes, positive immunoreactivity signals were visualized using a Nickel-DAB method [DAB Peroxidase (HRP) Substrate Kit (with Nickel), 3,3'-diaminobenzidine SK-4100, Vector Laboratories, Burlingame, CA, USA or Sigma-Aldrich, St. Louis, MO, USA].

## Primary cell culture

In order to ensure the detection of neuron specific p-ERK, we set up primary culture from neonatal mouse brain. Briefly, cortex isolated from neonatal mouse brain were mechanically minced by Castro-viejo scissor and trypsinized by Trypsin (SI-T4174-100ml, Thermo Fisher Scientific, Waltham, MA, USA) diluted in Hanks Buffered Salt Solution (HBSS) (SI-H6648-500ml, Thermo Fisher Scientific, Waltham, MA, USA) with L-glutamine (2 mM/ml) (SI-G7513-100ml, Thermo Fisher Scientific, Waltham, MA, USA) for 15 min at 37 °C with three subsequent HBSS washes before treated by deoxyribosenuclease I (SI-D4513-1vl, Thermo Fisher Scientific, Waltham, MA, USA). The treated tissues were then triturated with fire polished glass pipette and strained through 40 μm strainer (431750, Corning Inc, Corning, NY, USA) and seeded on plasma treated and poly-D-Lysine (SI-P7405 Thermo Fisher Scientific, Waltham, MA, USA) coated glass cover slips at a density of $10^5$ /ml in B27+ supplemented (Gibco A3582801, Thermo Fisher Scientific, Waltham, MA, USA) neurobasal media (Gibco A3582901, Thermo Fisher Scientific, Waltham, MA, USA) with 10 % horse serum (Gibco 26050070, Thermo Fisher Scientific, Waltham, MA, USA) and penicillin/streptomycin (100 U/ml) (Life Technologies, Carlsbad, CA, USA). Culture was gradually replaced with serum free B27+ neurobasal media until day seven for either immunofluorescence or for live cell calcium signal detection.

## Live cell calcium signal imaging

In order to visualize calcium signal in the neurites and in the cell bodies of neuron, we treated the primary cultures with three different green fluorescent dyes, that is Invitrogen Oregon Green 488 BAPTA-1, AM cell permeant (O6807, Thermo Fisher Scientific, Waltham, MA, USA), Invitrogen Fluo-4, AM, FluoroPure grade (F23917, Thermo Fisher Scientific Waltham, MA, USA), or Invitrogen Fura-2, AM, cell permeant (1 mM Solution in Anhydrous DMSO) (F1225, Thermo Fisher Scientific Waltham, MA, USA). Living primary culture on cover slip was immersed in HHBS (20 mM Hepes pH7.4, 1 mM $CaCl_2$, 0.5 mM $MgCl_2$, 0.4 mM $MgSO_4$-$7H_2O$, 5 mM KCl, 0.4 mM $KH_2PO_4$, 4 mM $NaHCO_3$, 138 mM NaCl, 0.3 mM $Na_2HPO_4$, 6 mM D-Glucose) with 2–5 μM of fluorescent dye and incubate in incubator for 90 min. Subsequently, calcium staining solution was replaced with HHBS with 17 % neurobasal media. Cover glass was mounted on an imaging chamber and placed under the fluorescent micro-scope and micropipette ultrasound was set up to the proximity of targeted cells.

Images were recorded using Olympus IX71 fluorescent microscope (Olympus Corporation, Shin-juku, Tokyo, Japan) with digital camera for microscope Camera attachment with 0.63 x lens (DP80, Olympus Corporation, Shinjuku, Tokyo, Japan). Stacked images were analyzed in ImageJ. ROI of neurites or cell bodies were determined for stacks resliced to obtain data of fluorescence intensities plotted against time points.

## Molecular signaling protein inhibitors

To determine whether PIEZO receptor or TRPC1 was responsible for the signal, we applied the GsMTx-4 (500 nM) (*Bae et al., 2011*; *Bowman et al., 2007*) (ab141871, Abcam Inc, Cambridge, MA, USA) isolated from tarantula venom and Gadolinium (10 μM) (*Coste et al., 2010*) (G7532, Sigma-Aldrich, St. Louis, MO, USA) to the tissues or cells before ultrasound treatment. To investigate the potential role of ASIC channels in ultrasound signal transduction, we utilized the inhibitors such as Amiloride (100 μM) (*Leng et al., 2016*) (A7410-1G Sigma-Aldrich, St. Louis, MO, USA) and PcTx1 (0.1–50 nM) (*Cristofori-Armstrong et al., 2019*) (Tocris #5042, Bio-Techne Corporation, Minneap-olis, MN, USA). To test whether endoplasmic reticular stored calcium was involved in the calcium signal detected, RyR inhibitor, JTV519 fumarate (10 μM) (*Hunt et al., 2007*) (Tocris #4564, Bio-Techne Corporation, Minneapolis, MN, USA) and Thapsigargin (T9033, Sigma-Aldrich, St. Louis, MO, USA) was tested.

## Immunofluorescence staining of DCX

After the VLIUS stimulation, the mice were sacrificed and perfused with 10 % formaldehyde/PBS. The brain was then harvested and fixed with 10 % formaldehyde/PBS at room temperature. Samples were embedded in paraffin and serial 7 μm transverse sections were mounted on slides. The samples were deparaffinized, rehydrated, antigen retrieved (100°C, 20 min) and washed in PBST. Slices were blocked with 10 % newborn calf serum (NCS) and 1 % BSA in PBST for 1 hr, incubated with primary antibody overnight at 4°C. After washing with PBST, the samples were incubated with the secondary antibody

for 1 hr at room temperature, washed with PBST and mounted with EverBrite Hardset Mounting Medium containing DAPI to label the nuclei (Biotium). Slides were viewed, and images were captured with LSM780 confocal microscope (Zeiss, Jena, Germany). The primary antibodies used for immunostaining and their dilutions were as follows: rabbit anti-DCX (1:200, Cell signaling), mouse anti-MAP2 (1:200, Thermo). The secondary antibodies used were Alexa Fluor 488-conjugated goat anti-rabbit IgG (1:100, Thermo) and Alexa Fluor 555-conjugated goat anti-mouse IgG (1:100, Thermo).

## Data and statistical analyses

Cells were counted using ImageJ. Cells were identified using a global threshold with watershed segmentation. The number of pixel groups was evaluated as the number of cells. Cells were also manually counted from bright-field images. Measurements were compared between control and ultrasound groups using student t-test. A p value ≤ 0.05 was considered to indicate statistical significance. All statistical analyses of animal studies were performed using GraphPad Prism 8.

## Acknowledgements

This study was supported by Ministry of Science and Technology, Taiwan (MOST107-2221-E-002-068-MY3, MOST108-2321-B-002-047, MOST 108-2321-B-002-061-MY2), National Health Research Institute, Taiwan (NHRI-EX109-10924EI), National Taiwan University (NTU-CC-107L891105); and grants from MOST, Taiwan (MOST 108-2321-B-001-028-MY2, MOST 110-2321-B-001-010) to CCC.

## Additional information

### Funding

| Funder | Grant reference number | Author |
|---|---|---|
| Ministry of Science and Technology, Taiwan | MOST 107-2221-E-002-068-MY3 | Jaw-Lin Wang |
| Ministry of Science and Technology, Taiwan | MOST108-2321-B-002-047 | Jaw-Lin Wang |
| Ministry of Science and Technology, Taiwan | MOST 108-2321-B-002-061-MY2 | Jaw-Lin Wang |
| National Health Research Institutes | NHRI-EX109-10924EI | Jaw-Lin Wang |
| Ministry of Science and Technology, Taiwan | MOST 108-2321-B-001-028-MY2 | Chih-Cheng Chen |
| Ministry of Science and Technology, Taiwan | MOST 110-2321-B001-010 | Chih-Cheng Chen |
| National Taiwan University | NTU-CC-107L891105 | Jaw-Lin Wang |

The funders had no role in study design, data collection and interpretation, or the decision to submit the work for publication.

### Author contributions

Jormay Lim, Conceptualization, Data curation, Formal analysis, Investigation, Methodology, Project administration, Supervision, Validation, Writing - original draft, Writing – review and editing; Hsiao-Hsin Tai, Data curation, Formal analysis, Investigation, Methodology, Validation; Wei-Hao Liao, Chen-Ming Hao, Data curation, Formal analysis, Investigation, Methodology; Ya-Cherng Chu, Conceptualization, Data curation, Validation, Writing – review and editing; Yueh-Chun Huang, Shao-Shien Lin, Data curation, Formal analysis, Methodology, Validation; Cheng-Han Lee, Animal experiments, Methodology; Sherry Hsu, Data curation, Formal analysis, Investigation; Ya-Chih Chien, Methodology; Dar-Ming Lai, Wen-Shiang Chen, Funding acquisition, Resources, Supervision; Chih-Cheng Chen, Conceptualization, Methodology, Supervision, Visualization, Writing – review and editing; Jaw-Lin Wang, Conceptualization, Funding acquisition, Project administration, Resources, Supervision, Writing – review and editing

## Author ORCIDs

Jormay Lim (iD) http://orcid.org/0000-0001-7191-545X
Ya-Cherng Chu (iD) http://orcid.org/0000-0002-6408-2813
Jaw-Lin Wang (iD) http://orcid.org/0000-0002-5734-9276

## Ethics

The animal work has been performed following the recommendations in the Guide for the Care and Use of Laboratory Animals of National Taiwan University. All of the animals handling complies to the IACUC protocol #20190055 of National Taiwan University Hospital.

## Decision letter and Author response

Decision letter https://doi.org/10.7554/eLife.61660.sa1
Author response https://doi.org/10.7554/eLife.61660.sa2

## Additional files

### Supplementary files

- Transparent reporting form
- Supplementary file 1. Supplementary files 1A-H.

### Data availability

All data generated relevant to this study are included in the manuscript are presented in the manuscript either as main figures or as figure supplements. Source data files will be provided when there is a need.

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

## Appendix 1

**Appendix 1—key resources table**

| Reagent type (species) or resource | Designation | Source or reference | Identifiers | Additional information |
|---|---|---|---|---|
| Other | Wildtype C57BL/6 J (*Mus musculus*) | The Jackson Laboratory | Stock number: 000664 | Experimental Animal Facility, Institute of Biomedical Sciences, Academia Sinica IACUC 12-03-332, 20-06-1492 |
| Other | *Asic1a$^{-/-}$* (specific targeting alternative spliced isoform *Asic1a$^{-/-}$*) C57BL6/J (*Mus musculus*) | Eur J Neurosci 2015 Jun; 41(12):1553–68. doi:10.1111/ejn.12905. | *Asic1$^{-/-}$* | Experimental Animal Facility, Institute of Biomedical Sciences, Academia Sinica IACUC 12-03-332, 20-06-1492 |
| Other | *Asic3$^{-/-}$* C57BL6/J (*Mus musculus*) | Nat Commun 2016 May; 7:11,460. doi:10.1038/ncomms11460. | *Asic3$^{-/-}$* | Experimental Animal Facility, Institute of Biomedical Sciences, Academia Sinica IACUC 12-03-332, 20-06-1492 |
| cell line (*Cricetulus griseus*) | Epithelial like Chinese Hamster ovary cells | ATCC | CCL-61 (RRID:CVCL_0214) | |
| Transfected construct (*Discosoma sp.*) | pCMV - mCherry | Gift from Dr. Huang, Yi-Shuian, IBMS, Academia Sinica, Taiwan | Vector control | 1 µg DNA: 3 µl Lipofectamin 2000 in 1 ml opti-MEM medium |
| Transfected construct (*Mus musculus*) | pT7-RFP-N2- T▲T - P2A - mASIC1a (alternative spliced isoform *Asic1a*) | The cDNA of mouse ASIC1a was subcloned into the plasmid (pT7-RFP-N2) | *Asic1* | 1 µg DNA: 3 µl Lipofectamin 2000 in 1 ml opti-MEM medium |
| Antibody | pERK; Rabbit polyclonal [polyclonal Phospho-p44/42 MAPK (ERK1/2) (Thr202/Tyr204)] | Cell Signaling Technology | #9,101 (RRID:AB_331646) | IHC (1:500) |
| Antibody | Doublecortin; (Rabbit polyclonal) | Cell Signaling | #4,604 (RRID:AB_561007) | ICC (1:200) |
| Antibody | MAP2; (Mouse monoclonal) | Thermo Fisher Scientific | MA5-12823 (RRID:AB_10982160) | ICC (1:200) |
| Antibody | Secondary biotinylated goat-anti-rabbit antibodies; (Goat monoclonal) | Vector Laboratories | BA-1000–1.5 (RRID:AB_2313606) | (1:1000) |
| Antibody | Alexa Fluor 488-conjugated goat anti-rabbit IgG; (Goat monoclonal) | Thermo Fisher Scientific | 16–237 (RRID:AB_436053) | (1:100) |
| Antibody | Alexa Fluor 555-conjugated goat anti-mouse IgG; (Goat monoclonal) | Thermo Fisher Scientific | A-21422 (RRID:AB_141822) | (1:100) |
| Commercial assay or kit | IHC Reagent; Avidin-Biotin pre-mix solution | Vector Laboratories | A-2004–5 (RRID:AB_2336507) | (1:200) |
| Commercial assay or kit | IHC staining kit; DAB Peroxidase Substrate Kit with Nickel, 3,3'-diaminobenzidine | Vector Laboratories | SK-4100 (RRID:AB_2336382) | According to instruction manual |

*Appendix 1 Continued on next page*

*Appendix 1 Continued*

| Reagent type (species) or resource | Designation | Source or reference | Identifiers | Additional information |
|---|---|---|---|---|
| Commercial assay or kit | Live cell imaging staining reagent for calcium; Invitrogen Oregon Green 488 BAPTA-1, AM, cell permeant | Thermo Fisher Scientific | O6807 | 5 µM |
| Commercial assay or kit | Live cell imaging staining reagent for calcium; Invitrogen Fluo-4, AM, FluoroPure | Thermo Fisher Scientific | F23917 | 5 µM |
| Commercial assay or kit | Live cell imaging staining reagent for calcium; Fura-2, AM, cell permeant | Thermo Fisher Scientific | F1221 | 5 µM |
| Chemical compound, drug | GsMTx-4 | Abcam Inc | ab141871 | 500 nM |
| Chemical compound, drug | Amiloride | Sigma-Aldrich | A7410-1G | 100 µM |
| Chemical compound, drug | PcTx1 | Tocris | #5,042 | 0.1–50 nM |
| Chemical compound, drug | Ruthenium Red | Tocris | #1,439 | 5–10 µM |
| Chemical compound, drug | RyR inhibitor, JTV519 fumarate | Tocris | #4,564 | 10 µM |
| Chemical compound, drug | Thapsigargin | Sigma-Aldrich | T9033 | 100 nM |
| Chemical compound, drug | Cytochalasin D | Tocris | #1,233 | 5 µg- 10 µg/ml |
| Chemical compound, drug | Nocodazole | Tocris | #1,228 | 5 µg- 10 µg/ml |
| Software, algorithm | Image J | Abràmoff, M. D., Magalhães, P. J., & Ram, S. J. (2004). Image processing with ImageJ. *Biophotonics international*, *11*(7), 36–42. | Particle analysis | Set threshold and define particle sizes |
| Software, algorithm | GraphPad Prism 8 | Swift, M. L. (1997). GraphPad prism, data analysis, and scientific graphing. *Journal of chemical information and computer sciences*, *37*(2), 411–412. | Student t-test | |

