## [Decision Letter]

**Acceptance summary:**

This study reports a novel role for the mechanosensitive trimeric cation channel ASIC1 in activation of neurons of the mouse brain by ultrasound stimulation.

**Decision letter after peer review:**

Thank you for submitting your article "ASIC1a is required for neuronal activation via low-intensity ultrasound stimulation in mouse brain" for consideration by *eLife*. Your article has been reviewed by 2 peer reviewers, and the evaluation has been overseen by a Reviewing Editor and Andrew King as the Senior Editor. The reviewers have opted to remain anonymous.

The reviewers have discussed the reviews with one another and the Reviewing Editor has drafted this decision to help you prepare a revised submission.

As the editors have judged that your manuscript is of interest, but as described below that additional experiments are required before it can be considered further for publication, we would like to draw your attention to changes in our revision policy that we have made in response to COVID-19 (https://elifesciences.org/articles/57162). First, because many researchers have temporarily lost access to the labs, we will give authors as much time as they need to submit revised manuscripts. We are also offering, if you choose, to post the manuscript to bioRxiv (if it is not already there) along with this decision letter and a formal designation that the manuscript is "in revision at *eLife*". Please let us know if you would like to pursue this option. (If your work is more suitable for medRxiv, you will need to post the preprint yourself, as the mechanisms for us to do so are still in development.)

Summary:

This is an interesting manuscript suggesting that ultrasound stimuli induce movements of the extracellular matrix and the cytoskeleton to cause mechanical activation of ASIC1a in cortical neurons. This is a novel finding.

Essential revisions:

The reviewers find the results of the study to be intriguing, but have also voiced major concerns about open questions and methodological issues. Most importantly, the reviewers felt that there is insufficient evidence to support a key role for ASIC1a in the responses to ultrasound stimulation.

A revised version would need to address and include the following:

1. It is critically important to back up the claim that ASIC1a mediates the effects described in both in vivo and in vitro experiments. For in vivo experiments pertaining to pERK activation by ultrasound stimulation, appropriate ASIC knockout mice should be used. This experiment is critical for the manuscript to be considered further for a re-review. For in vitro experiments in HEK cells, the reviewers wish to see transfection with ASIC1a and tests performed to determine whether this is sufficient to confer sensitivity to ultrasound stimulation.

2. Calcium imagine experiments: Number of cells tested in calcium imaging experiments must be specified in legends, it should be clarified whether data points refer to individual cells or to mean values, and are sufficient enough for robust interpretations. Data on internal calibration and integrity of cellular responses, e.g. using an ionophore, should be shown. You should clarify if the inhibitors were applied to the control cells from the same panel or to different cells. Please specify how many control cells actually responded to the ultrasound stimulation.

[Editors' note: further revisions were suggested prior to acceptance, as described below.]

Thank you for submitting your article "ASIC1a is required for neuronal activation via low-intensity ultrasound stimulation in mouse brain" for consideration by *eLife*. Your article has been reviewed by 1 peer reviewer, and the evaluation has been overseen by a Reviewing Editor and Andrew King as the Senior Editor. The following individual involved in review of your submission has agreed to reveal their identity: Gary R Lewin (Reviewer #3).

Essential revisions:

1. Revise the text to avoid any claim that activation of ASIC1 channel is sufficient for neuronal activation by ultrasound stimulation and that this is necessarily linked to mechanical activation of the channel. Please discuss the alternative scenarios via which ASIC3 could be contributing to this effect.

2. Correct the assertion that ASIC1 is gated via tethers.

3. Ensure that all the images and photomicrographs have scale bars.

*Reviewer #3:*

The authors have made a very significant effort to add new data to the manuscript. Most noteworthy is the new analysis of ultrasound induced neuron activation (pERK as a surrogate) in both ASIC1 and ASIC3 knockout mouse models. The authors noted a higher baseline number of pERK positive cells in ASIC1-/- mice but no increase was seen after ultrasound exposure. This data together with the data from ASIC3 mutant mice where no attenuation of the ultrasound effect was seen, better support a role for ASIC1 in the ultrasound effect. The authors should still note that this data implicates a role for ASIC1 in being necessary for the effect, but does not show that the mechanical activation of this trimeric channel is the mechanism accounting for the in vivo effect. For example the presence of ASIC1 could be necessary for the trafficking or functionality of other proteins necessary for the effect. Indeed the authors should at least comment in the discussion on the fact that ASIC1a is highly selective for sodium ions cannot directly account for calcium influx measured in the in vitro experiments.

One other general point that the authors should take care of is their assertion that ASIC1a is gated via tethers. As far as I am aware there is no direct evidence of a molecular tether that binds to this protein to gate the channel. Rather, this assertion is based on the observation that indentation of the substrate can activate a such channels (work from the Chen lab). However, this has also been shown for PIEZO and TRPV4 channels PMID: 28135189,PMID: 24662763. These data were reviewed in a short review here PMID: 24981693.

---

## [Author Response]

Reviewer #1:In the manuscript entitled “ASIC1a is required for neuronal activation via low-intensity ultrasound stimulation in mouse brain", Lim et al. investigate the mechanism underlying the activation of brain neurons by transcranial low-intensity ultrasound stimulation. The authors propose that ultrasound stimuli-induced movements of the extracellular matrix and the cytoskeleton cause mechanical activation of ASIC1a in cortical neurons, which leads to ca^2+^ influx and subsequent expression of pERK, which the authors used as a surrogate marker for neuronal activation.While I agree that the finding that ultrasound activates neurons via activation of a mechanosensitive ion channel is per se very interesting, I have to say that in my opinion most of the conclusions and claims are not supported by the actual data.1. The entire study is purely correlative. Thus, the authors made two independent experiments; on the one hand they show that in-vivo transcranial ultrasound stimulation induces pERK in various brain regions and on the other hand they shown that ultrasound-evoked ca^2+^ influx in cultures of cortical neurons is probably mediated by ASIC1a. From this data they conclude that pERK activation is also mediated by ASIC1a activation. This is, however, pure speculation. The authors must provide additional evidence to support their claim. In my opinion the sole use of PcTx1 is not sufficient to prove that the ca^2+^ signals are mediated by ASIC1a. Hence, firstly the authors should demonstrate that ASIC1a is indeed activated by ultrasound. This is a very simple experiment. All they would have to do is express ASIC1a in a cell line (e.g. HEK293, CHO, etc) and show that this expression renders the cells sensitive to ultrasound. Second, I would appreciate it if the authors would show that cortical neurons, especially those that show pERK activation, express ASIC1a in the first place. This would also be quite simple – just co-stain the brain sections with an anti-ASIC1a antibody. Third, if the authors want to keep up their claim (see title) that ASIC1a is required for ultrasound activation of brain neurons they should examine ultrasound-induced pERK activation in ASIC1a-knockout mice.

We have performed the *Asic1a* overexpression experiment suggested by the reviewer using CHO cells. The CHO cells are in favour instead of HEK293 because of the endogenous ASIC1a expression levels in HEK293. The results are organized into Manuscript Figure 4. (Unless otherwise indicated, all the figures are referred to according to the manuscript figure number system). Basically, our results confirm the notion that ASIC1a is mediating the neuronal calcium response. The CHO cells calcium response to pipette delivered ultrasound with a time lag of about 10-15 seconds (Figure 4*A*,*B*). We think this may be due to acoustic flow however the delay response is not within our scope of study. The CHO cells overexpressing *Asic1a* show an immediate calcium response to ultrasound (Figure 4 *A*, *B*) comparable to that was detected in primary cultured neurons (Manuscript Figure 2). In order to make sure that the difference is not due to the mere effect to an introduction of the ectopic DNA into the cells, same experiments were conducted using a vector transfected control to compare with *Asic1a* transfected cells. We have observed and confirmed that the shortened response time to the ultrasound stimulation is due to ectopic ASIC1a (Figure 4*C*). The responsive time was significantly reduced in *Asic1a* overexpressed CHO cells (Figure 4 *C*). After the stimulations, intracellular calcium can be elevated again by 0.1% saponin treatment to cause cell permeation (Figure 4*D*). The calcium response calibrated by 0.01% Saponin cell perforation could surge to the maximal Fura-2 ratio (F_340/380nm_) (Figure 4—figure supplement 1 *C, D*). The 2-way ANOVA analysis of this set of data showed that ASIC1a overexpression did not contribute to the difference of with an insignificant *p* value (F=0.11; p=0.74) (Figure 4-table supplement 2) while cell perforation contributed significantly to the calcium responses (F=10.52; p<0.0001) (Figure 4-table supplement 2). Note that the calcium response upon ultrasound treatment remained effective under the PcTx1 treatments in vehicle transfected cells (Figure 4*E*) as ultrasound as a factor still contributed significantly (F=17.2; p<0.0001) to the changes of calcium while there was no significant interaction (F=1.44; p=0.15) (Figure 4-table supplement 3) of drug treatments with ultrasound stimulations (Figure 4-table supplement 3). In contrast, when CHO cells were overexpressing ASIC1a, ultrasound stimulations had no significant effects on the calcium response under the PcTx1 inhibition (Figure 4*F*) (F=1.26; p=0.29) (Figure 4-table supplement 4), while the two factors interact significantly (F=4.46, p<0.0001) (Figure 4-table supplement 4). In short, different from the failed PcTx1 inhibition of mock control (Figure 4 *E*), the ASIC1a mediated calcium response upon ultrasound was significantly blocked by PcTx1 starting from as diluted as 2nM (Figure 4*F*).

The reviewer ask for the immunostaining of ASIC1a expression in the cortical neurons. However, the immunostaining of ASIC1a is currently not possible due to the lack of reliable antibody. We have tried many commercially available antibodies using *Asic1a^-/-^* to validate the specificity however we have found no antibody that detect specific signal that is absent in knockout samples.

Despite the co-staining of ASIC1a is not possible because there is no good and specific antibody available that can meet the standard of knockout validation. Nevertheless, we have obtained three batches of *Asic1a^-/-^* mice and performed pERK immunohistochemical staining. The results confirmed that the deletion of *Asic1a* in mice can sufficiently suppressed pERK cell count in cortex, hippocampus and amygdala Figure 7, demonstrating that ASIC1a is required for ultrasound activation of cells in mouse brain. The two-way ANOVA analysis of p-ERK cell counts showed that there was only an interaction of two factors, namely genotype and ultrasound treatment in the cortex (F=6.45, p=0.0037) but not in hippocampus and amygdala (Figure 7-table supplement 1). To further confirm the specific function of ASIC1a in mediating ultrasound stimulation, we included the *Asic3^-/-^* in our p-ERK response phenotypes studies and indeed the genotype did not reduce the activation of p-ERK as *Asic1a^-/-^* did. These results indicated that the p-ERK response in mouse brain is likely directly caused by the transcranial ultrasound instead of caused by the secondary effects due to the neurons wired with auditory circuits or other sensory circuits, as ASIC3 being mainly expressed in somatosensory neurons and spiral ganglion neurons (1, 2).

2. It is difficult to evaluate the ca^2+^ imaging experiments, because the method – especially the ultrasound stimulation – is not very well described. Hence it is unclear to me how close to the cell the ultrasound stimulator was placed. Moreover, the N-numbers of the ca^2+^ imaging experiments are rather small (by the way, it would make reading much easier if the N-numbers were indicated in the figure). Most importantly, it is unclear if the inhibitors (Gadolinium, GsMTx4 etc – Figure 2B-H) were applied to the control cells from the same panel or to different cells. In this context it would be important to know how many control cells actually responded to the ultrasound stimulation. Considering the low N-number, I was wondering if the authors may have had a hard time finding cells that responded and that this is the reason why the N-numbers are so small? I suggest examining many more control neurons and provide information about the proportion of cells that respond. If only for the controls as well as for the cells treated the various channel inhibitors.

To address the concern of the validity of the ca^2+^ imaging experiments and to allow better evaluation of calcium response, we have performed the experiment again using Fura-2 staining reagent and setting up the imaging system recording F_340/380nm_ images. The results were shown in (Figure 2—figure supplement 2 and Video ).

The same cells were stimulated for the second time and still show a good response within the duration of ultrasound stimulation (Figure 2—figure supplement 2). It is probably due to the imaging system in our laboratory, the photo-decay problem is quite overwhelming, and it will be difficult for us to keep recording the same visual field for subsequent treatments. Thus, although the inhibitors were applied to the same dish, a different visual field is recorded for the quantification of the calcium response. The N-numbers presented in our manuscript is representing a subset of what we have tested, mainly from the experiments using Invitrogen Oregon Green 488 BAPTA-1, AM, cell permeant. When this project was funded year 2019, the Fura-2 staining method and imaging system was not available in our lab yet. When we used Oregon Green 488 BAPTA-1, almost all of the tested visual fields (>90%) were responsive to ultrasound stimulation. Thus, this allowed a better consistency in quantification. On the other hand, if Invitrogen Fluo-4, AM, FluoroPure was used, only 28.6%-38.5% of the tested visual fields were showing the calcium response upon ultrasound stimulation (Video 4). When this staining method is applied, each treatment is performed using a fresh dish of cells. After the image recording, we selected the best responding cells from each experimental group for comparison to the sham-treated control dish of cells. The N-numbers of every graph and chart presented are added to the manuscript figure legends.

Figure 2—figure supplement 1C shows the position of micropipette in the visual field of video 3.

Reviewer #2:In this study the authors claim that short lasting low intensity ultrasound stimulation activates many neurons in the whole brain. They further claim that the activation mechanism is via the ASIC1a channel. There are some intriguing results in this paper, but there are also many open questions and methodological issues that should be addressed. The authors use pERK as a surrogate for neuronal activation by a global ultrasound stimulus. Some but not all neurons in cortex seem to show activation (it seems only large pyramidal cells, why not interneurons? More analysis needed here).

We have tried to identify the p-ERK cells by co-staining them with several markers, including NeuN, NMDAR, GAD67 and PV. The results in shown in the following micrographs Figure 7—figure supplement 1. We observed that there were visibly a large number of p-ERK co-stained with NeuN (94% or 197/209). However, we have not found any NMDAR co-stained cells in our experiment. We cautioned that this could be a false negative result that was probably due to the limitation of antibody specificities and staining procedures. We found that there is very small population of p-ERK positive cells co-stained with GAD67 (4.5% or 10/223) while even smaller percentage co-stained cells of p-ERK with Parvalbumin (PV) (0.9% or 2/211).

This experiment is followed by an in vitro experiment with cultured cortical neurons from neonates (no ages given for the animals used in this experiment as far as I can see). These are also not equivalent to the adult cells tested in the in vivo experiment. In the bulk of the experiments calcium imaging is used as a surrogate to measure neuronal activation. Unfortunately, in none of the graphs displayed of the Δ F/Fo is there any indication of the number of cells selected and measured. This is critical to evaluate the robustness of the results. In addition, it is normal at the end of the experiment to permeabilize the neurons to calcium by using an ionophore. This allows the assessment of the maximum fluorescence signal when calcium outside concentration equilibrates with the intracellular concentration. This was not done which means the experiments have no internal calibration.

The neuronal cultures were prepared using postnatal day 3 – postnatal day 5 pups. The usage of neonatal primary culture is to have a proof-of-concept that ultrasound can directly activate neurons in a simplified in vitro system. Of note, it is almost impossible to culture adult cortical neurons for calcium imaging studies. To ensure that the results are reproducible and robust, we have applied four different staining reagents; namely (1) Invitrogen Oregon Green 488 BAPTA-1, AM, cell permeant, (2) Invitrogen Fluo-4, AM, FluoroPure, (3) Rhod-2 AM, fluorescent ca^2+^ indicator (ab142780) and (4) Fura-2, AM, cell permeant. We found that Rhod-2 AM is not compatible with our system and we cannot obtain any good signal from the staining. Thus, this staining method was ruled out. Our observation can be reproduced by using the other three reagents. Of note only that when Fluo-4 is applied, the responding neurons are about 28.6%-38.5% among tested (6/21 and 10/26, in total 46 cells were tested).

We have added 0.1% saponin to show the assessment of the maximum fluorescence signal. Representative results are shown in Figure 4—figure supplement 1.

Likewise, similar cell perforation experiments have been performed using Oregon Green 488 BAPTA-1 for primary neurons to show the maximal calcium response presented in DF/F0 (Figure 2—figure supplement 3). We have also measured the repeated stimulation of the same cells using this staining method and the comparisons of first and second stimulation is shown in Figure 2—figure supplement 3D.

It is for me impossible to assess the robustness of the calcium imaging experiment when I do not know what each data point corresponds to, take Figure 2I as an example. Are these individual cells or means values from many cells from individual cultures? Many critical methodological details are indeed missing from the paper.

Yes. Every data point corresponds to individual cell. Although we have performed the experiments for either two or three times for each treatment, we presented the data from a representative experiment because the staining reagents and imaging settings are sometimes changed, and it is quite impossible to merge all the measurements into one graph. To demonstrate that the calcium response is indeed reproducibly stimulated by the micropipette guided ultrasound and not caused by cell damage, we have set up a new imaging system in the lab to perform the experiment again using Fura-2 staining reagent and the quantification is presented by a ratio of F_340nm/380nm_. The results were shown in Figure 2—figure supplement 2 and Video 5.

The same cells were stimulated for the second time and still show a good response within the duration of ultrasound stimulation (Figure 2—figure supplement 2). The photo-decay problem causes a difficulty in keeping the same visual field for different treatments of inhibitors. Thus, although the inhibitors were applied to the same dish, a different visual field is recorded for the quantification of the calcium response. The N-numbers presented in our manuscript is representing a subset of what we have tested, mainly from the experiments using Invitrogen Oregon Green 488 BAPTA-1, AM, cell permeant. When we use this reagent, almost every visual field of the tested cells (>90%) were responsive to ultrasound stimulation (Video 3). We place the pipette right on top of the imaged neuron or its neurites under 40X objective lens, the calcium response can be observed at the time points of 1.5 to 5 seconds after the turning on of ultrasound. This can be because of the limitation of our recording setting, which is usually capturing 1 or 2 images per second. In most of the cases, about 1or 2 neuronal cells per visual field can be quantified plotting fluorescence (DF/F0) against time (s). On the other hand, if Invitrogen Fluo-4, AM, FluoroPure is used, only 28.6%-38.5% of the visual fields were showing calcium response upon ultrasound stimulation (Video 4). When this staining method is applied, each treatment is performed using a fresh dish of cells. After the image recording, we select the top responding visual fields from each experimental group for comparison to the control dish of cells.

The idea that ASIC1a is THE critical mediator of this effect is quite surprising and the more dramatic and implausible the conclusion may seem, the more solid the evidence needed. The authors should use ASIC1a mutant mice both in vivo and in vitro to prove that ASIC1a really is critical. The same applies to the apparent effect on neurogenesis.

We have prepared *Asic1a* gene deletion mice (*Asic1a^-/-^*) and *Asic3* gene deletion mice (*Asic3^-/-^*) for IHC experiments measuring p-ERK response to address the reviewer’s concern for whether *Asic1a* really play a major role mediating transcranial ultrasound stimulation in mice. Transcranial ultrasound induces p-ERK response in various regions of the mice including cortex, hippocampus and amygdala (representative micrographs Figure 7 *A, B, H, I, O, P*), while the activation is partially abolished in the *Asic1a^-/-^* (Figure 7 *C, D, J, K Q, R*). Therefore, the *Asic1a^-/-^* causes the p-ERK response upon transcranial ultrasound stimulation in the three regions of the brain to become not statistically significant (Figure 7 *G, N, U*). On the other hand, the effects of *Asic3^-/-^* on the p-ERK response to ultrasound are less noticeable (Figure 7 *E, F, L, M, S, T*). The ultrasound induced p-ERK response is therefore remaining statistically significant (Figure 7 *G, N, U*). The two-way ANOVA analysis of p-ERK cell counts showed that there was only an interaction of two factors, namely genotype and ultrasound treatment in the cortex (F=6.45, p=0.0037) but not in hippocampus and amygdala (Figure 7-table supplement 1). To further confirm the specific function of ASIC1a in mediating ultrasound stimulation, we included the *Asic3^-/-^* in our p-ERK response phenotypes studies and indeed the genotype did not reduce the activation of p-ERK as *Asic1a^-/-^* did. These results indicated that the p-ERK response in mouse brain is likely directly caused by the transcranial ultrasound instead of caused by the secondary effects due to the neurons wired with auditory circuits or other sensory circuits, as ASIC3 being mainly expressed in somatosensory neurons and spiral ganglion neurons (1, 2).

We have shown the increase levels of DCX caused by consecutive ultrasound stimulations. This phenomenon has been confirmed independently by two other labs of our collaborators using 4-weeks old mice although we have only reported data produced by the initial batch of mice of 6-weeks. And to address the reviewer’s concern, we have performed the same experiment again using mice with *Asic1a*^-/-^ (Figure 6).

The videos show quite large physical effects of the ultrasound on the cultures (cells moving around). This is problematic as it may be that what the calcium signals are purely indicative of cell damage. Controls should be provided to ensure this was not the case.

Previously our data were collected by staining reagent Invitrogen Oregon Green 488 BAPTA-1, AM, cell permeant. To ensure that the calcium response is not due to cell damage, we have performed an experiment with the ultrasound of input voltage of 400mVpp and Duty Factor 10% (Video 5). The cells tested under this condition are not moved while showing clear local calcium response with Fura-2 staining, which requires imaging system recording F_340nm/380nm_. We have stimulated the cells consecutively as shown in Figure 2—figure supplement 2.

Although there is obvious photo-decay after the first stimulation, the cells can recover after 5-10 minutes and be stimulated for the second time showing decent calcium response (Figure 2—figure supplement 2B). Similarly, the repeat stimulation of the same cells was evident in the Oregon Green 488 BAPTA-1-stained cells as shown in Figure 2—figure supplement 3D. In addition to the second round of stimulation despite overwhelming photo-bleaching effects, the calcium response of cell perforation due to the 0.01% Saponin treatment showed a different profile as presented in Figure 2—figure supplement 3C.

References

1. S. H. Lin *et al.*, Evidence for the involvement of ASIC3 in sensory mechanotransduction in proprioceptors. *Nat Commun*
**7**, 11460 (2016).

2. W. L. Wu, C. H. Wang, E. Y. Huang, C. C. Chen, Asic3(-/-) female mice with hearing deficit affects social development of pups. *Plos One* 4, e6508 (2009).

[Editors' note: further revisions were suggested prior to acceptance, as described below.]

Reviewer #3:The authors have made a very significant effort to add new data to the manuscript. Most noteworthy is the new analysis of ultrasound induced neuron activation (pERK as a surrogate) in both ASIC1 and ASIC3 knockout mouse models. The authors noted a higher baseline number of pERK positive cells in ASIC1-/- mice but no increase was seen after ultrasound exposure. This data together with the data from ASIC3 mutant mice where no attenuation of the ultrasound effect was seen, better support a role for ASIC1 in the ultrasound effect. The authors should still note that this data implicates a role for ASIC1 in being necessary for the effect, but does not show that the mechanical activation of this trimeric channel is the mechanism accounting for the in vivo effect. For example the presence of ASIC1 could be necessary for the trafficking or functionality of other proteins necessary for the effect. Indeed the authors should at least comment in the discussion on the fact that ASIC1a is highly selective for sodium ions cannot directly account for calcium influx measured in the in vitro experiments.One other general point that the authors should take care of is their assertion that ASIC1a is gated via tethers. As far as I am aware there is no direct evidence of a molecular tether that binds to this protein to gate the channel. Rather, this assertion is based on the observation that indentation of the substrate can activate a such channels (work from the Chen lab). However, this has also been shown for PIEZO and TRPV4 channels PMID: 28135189,PMID: 24662763. These data were reviewed in a short review here PMID: 24981693.

We thank the reviewer Prof Gary R Lewin and the editor Prof Rohini Kuner for your thorough examination of our works and your effort to help us refining our data interpretation and statements.

We have taken all the advices and altered our manuscript as listed below.

1. Original text line 47:

“ASIC1a and the tether-mode mechanotransduction were involved in the low-intensity ultrasound-mediated mechanotransduction and cultured neuron activation, which was inhibited by ASIC1a blockade and cytoskeleton-modified agents.”

Sentence altered to line 47:

“ASIC1a and cytoskeletal proteins were involved in the low-intensity ultrasound-mediated mechanotransduction and cultured neuron activation, which was inhibited by ASIC1a blockade and cytoskeleton-modified agents.”

2. Original text line 75-79:

Mechanosensitve ion channels are mainly grouped into bilayer model, such as PIEZO and TRP channels gated by membrane tension change, and extracellular matrix tethered model such as acid sensing ion channels (ASICs) (8-10). Here we aim to identify possible mechanical sensors in mouse brain that can directly respond to low-intensity ultrasound.

Sentences altered to line 77-80:

“Mechanosensitve ion channels such as PIEZO and TRP channels and acid sensing ion channels (ASICs) (8-10) are considered the candidates likely responsive to ultrasound. Here we aim to identify possible mechano-sensors in mouse brain that can respond to low-intensity ultrasound.”

3. Original text line 174-176:

Previous studies have shown ASICs are involved in tether mode mechanotransduction, which relies on intact cytoskeletal structures (8, 13).

Sentence altered to 184-186:

“Previous studies have suggested ASICs are involved in tether-mode mechanotransduction, which relies on intact cytoskeletal structures (8, 13).”

4. Original text line 180-183:

The collated data revealed a novel mode of ultrasound mechanotransduction with a combination of compression force and shear force that activates ASIC1a channels in mouse neurons via tether mode mechanotransduction (Figure 3 *F*).

Sentence altered to line 190-192:

“The collated data revealed a novel mode of ultrasound mechanotransduction with a combination of compression force and shear force that activates ASIC1a channels in mouse neurons (Figure 3 *F*).”

5. Addressing the issue of whether ASIC3 could be contributing to ultrasound induced changes.

Adding sentence to line 261-263:

“The inclusion of *Asic3^-/-^* is to elucidate the role of peripheral nerves in brain activation, as ASIC3 is highly expressed in in somatosensory neurons, trigeminal ganglion neurons, and spiral ganglion neurons.”

6. Original text line 259:

To further confirm the specific function of ASIC1a in mediating ultrasound stimulation,

Sentence altered to line 278:

“To test whether there is a role of peripheral nerves in mediating ultrasound stimulation,”

7. Addressing the issue of whether ASIC3 could be contributing to ultrasound induced changes.

Adding sentences to line 308-319:

“While the current view of transcranial ultrasound activation of neurons in brain is through the auditory nerves (22), our results from *Asic3^-/-^* mice suggest that the peripheral nerves may not play a role in the activation of p-ERK in mouse brain by low intensity ultrasound. Alternatively, the low intensity ultrasound-mediated mechanotransduction may act via a channel subtype-dependent manner specific for ASIC1a but not for other ASIC subtypes as shown in dextrose prolotherapy (23).”

8. Original text line 306:

To explain how ultrasound could activate ASIC1a via the tether-mode mechanotransduction,

Sentence altered to line 341:

“To explain the mode of ultrasound induced ASIC1a mechanotransduction,”

9. Original text line 309-310:

Considering the tether model in vitro,

Sentence altered to line 348-349:

“Considering the combinatorial forces mode in vitro,”

10. Addressing the issue of ASIC1a is functioning as a sodium channel, not calcium channel.

Adding sentence to line 353-355:

“As such, mechano-signal triggered ASIC1a, essentially a sodium channel, results in the intracellular calcium elevation possibly by activating voltage-gated calcium channels (30).”

11. Original text line 314-320:

The anchor-pulling condition in vivo on the other hand (Figure 3—figure supplement 1), is accomplished differently. Neurons are embedded in extracellular matrixes (earthy yellow color) such as laminin, poly-lysine, or poly-ornithine in the brain. When ultrasound is applied to the brain, extracellular matrixes anchor the ASIC1a (earthy arrow) while cytoskeletal changes pull it in a different direction (purple arrow), causing an activation of the cells manifesting as ERK phosphorylation.

Sentences altered to line 356-369:

“The condition in vivo on the other hand (Figure 3—figure supplement 1), is accomplished differently. Neurons are embedded in extracellular matrixes (earthy yellow color) such as laminin, poly-lysine, or poly-ornithine in the brain. ASIC1a is N-glycosylated at N366 and N393, both residues extracellular located (31). While N-glycosylation is reported to be involved in the surface trafficking and dendritic spine trafficking of ASIC1 (31, 32), the N-glycosylation of many proteins has been known to be important for adhesion and migration (33-35), implicating the extracellular matrix interacting nature of N-glycans. When ultrasound is applied to the brain, the acoustic pressure exerted through extracellular matrixes, can possibly activate ASIC1a via a cytoskeletal dependent manner (Figure 3—figure supplement 1), in addition to other mechanosensitive machineries such as PIEZO and TRPV4 (36, 37), since these mechanoreceptors have all been proven to be triggered by indentation of substrates. Consequently, this leads to an activation of the cells manifesting as ERK phosphorylation.”

12. Original text line 328-331:

The low-intensity ultrasound can directly activate neurons via tether-mode mechanotransduction and ASIC1a, which provides a molecular basis for future development of ultrasound neuromodulation.

Sentence altered to line 384-386:

“The low-intensity ultrasound can directly activate neurons via ASIC1a, which provides a molecular basis for future development of ultrasound neuromodulation.”

13. Original text line 695-696:

A schematic tether mode mechanotransduction model for in vivo circumstance.

Sentence altered to line 785-786:

“A schematic mechanotransduction model for in vivo circumstance.”

14. We have added scale bar in the micrographs for the following figures:

Figure 2 figure supplement 2

Figure 5

Figure 6

Figure 7 figure supplement 1